# Oxidative Stress, Inflammatory, Angiogenic, and Apoptotic molecules in Proliferative Diabetic Retinopathy and Diabetic Macular Edema Patients

**DOI:** 10.3390/ijms24098227

**Published:** 2023-05-04

**Authors:** Irene Andrés-Blasco, Alex Gallego-Martínez, Ximena Machado, Javier Cruz-Espinosa, Salvatore Di Lauro, Ricardo Casaroli-Marano, Víctor Alegre-Ituarte, José Fernando Arévalo, María Dolores Pinazo-Durán

**Affiliations:** 1Cellular and Molecular Ophthalmo-Biology Group, Department of Surgery, Faculty of Medicine and Odontology, University of Valencia, 46010 Valencia, Spain; iranbla@gmail.com (I.A.-B.); alexgamar@icloud.com (A.G.-M.); ximena199671@hotmail.com (X.M.); vicaleitu@gmail.com (V.A.-I.); 2Ophthalmic Research Unit “Santiago Grisolía”/FISABIO, 46017 Valencia, Spain; 3Spanish Net of Inflammatory Diseases and Immunopathology of Organs and Systems (REI/RICORS), Institute of Health Carlos III, Ministry of Science and Innovation, 28029 Madrid, Spain; rcasaroli@ub.edu (R.C.-M.); arevalojf2020@gmail.com (J.F.A.); 4Ophthalmology Department, Punta de Europa Hospital, 11207 Algeciras, Spain; fjcruzespinosa@gmail.com; 5Department of Ophthalmology, University Clinic Hospital, 47003 Valladolid, Spain; sadilauro@live.it; 6Spanish Net of Ophthalmic Pathology Research OFTARED, Institute of Health Carlos III, Ministry of Science and Innovation, 28029 Madrid, Spain; 7Department of Ophthalmology, Clinic Hospital, 08036 Barcelona, Spain; 8Department of Ophthalmology, University Hospital Dr. Peset, 46017 Valencia, Spain; 9Wilmer at Johns Hopkins Bayview Medical Center, Baltimore, MA 21224, USA

**Keywords:** proliferative diabetic retinopathy, vitreous body, blood, inflammation, angiogenesis, apoptosis, oxidative stress

## Abstract

The aim of this study is to evaluate molecules involved in oxidative stress (OS), inflammation, angiogenesis, and apoptosis, and discern which of these are more likely to be implicated in proliferative diabetic retinopathy (PDR) and diabetic macular edema (DME) by investigating the correlation between them in the plasma (PLS) and vitreous body (VIT), as well as examining data obtained from ophthalmological examinations. Type 2 diabetic (T2DM) patients with PDR/DME (PDRG/DMEG; n = 112) and non-DM subjects as the surrogate controls (SCG n = 48) were selected according to the inclusion/exclusion criteria and programming for vitrectomy, either due to having PDR/DME or macular hole (MH)/epiretinal membrane (ERM)/rhegmatogenous retinal detachment. Blood samples were collected and processed to determine the glycemic profile, total cholesterol, and C reactive protein, as well as the malondialdehyde (MDA), 4-hydroxynonenal (4HNE), superoxide dismutase (SOD), and catalase (CAT) levels and total antioxidant capacity (TAC). In addition, interleukin 6 (IL6), vascular endothelial growth factor (VEGF), and caspase 3 (CAS3) were assayed. The VITs were collected and processed to measure the expression levels of all the abovementioned molecules. Statistical analyses were conducted using the R Core Team (2022) program, including group comparisons and correlation analyses. Compared with the SCG, our findings support the presence of molecules involved in OS, inflammation, angiogenesis, and apoptosis in the PLS and VIT samples from T2DM. In PLS from PDRG, there was a decrease in the antioxidant load (*p* < 0.001) and an increase in pro-angiogenic molecules (*p* < 0.001), but an increase in pro-oxidants (*p* < 0.001) and a decline in antioxidants (*p* < 0.001) intravitreally. In PLS from DMEG, pro-oxidants and pro-inflammatory molecules were augmented (*p* < 0.001) and the antioxidant capacity diminished (*p* < 0.001), but the pro-oxidants increased (*p* < 0.001) and antioxidants decreased (*p* < 0.001) intravitreally. Furthermore, we found a positive correlation between the PLS-CAT and the VIT-SOD levels (rho = 0.5; *p* < 0.01) in PDRG, and a negative correlation between the PSD-4HNE and the VIT-TAC levels (rho = 0.5; *p* < 0.01) in DMEG. Integrative data of retinal imaging variables showed a positive correlation between the central subfield foveal thickness (CSFT) and the VIT-SOD levels (rho = 0.5; *p* < 0.01), and a negative correlation between the CSFT and the VIT-4HNE levels (rho = 0.4; *p* < 0.01) in PDRG. In DMEG, the CSFT displayed a negative correlation with the VIT-CAT (rho = 0.5; *p* < 0.01). Exploring the relationship of the abovementioned potential biomarkers between PLS and VIT may help detecting early molecular changes in PDR/DME, which can be used to identify patients at high risk of progression, as well as to monitor therapeutic outcomes in the diabetic retina.

## 1. Introduction

Currently, diabetes mellitus (DM) is a pandemic. The number of people with DM is expected to sharply increase to 643 million by 2030 and projected to rise to 783 million by 2045, representing 10% of the global population [1,2]. Key global findings strongly suggest that 541 million adults are at higher risk of developing type 2 DM (T2DM), constituting about 85–95% of all diabetics [2]. The DM pandemic has resulted in specific cardiovascular, renal, neurologic, and ocular complications, becoming the most frequent causes of end-stage disease worldwide. 

From an ophthalmologic viewpoint, diabetic retinopathy (DR) is detected in the ocular fundus examination. In some cases, it is necessary to perform fluorescein angiography, and for the evaluation of macular edema (DME), the use of optical coherence tomography (OCT) and OCT angiography (OCTA) is essential. The DR is a microangiopathic complication that affects patients, mainly in their working age, with any DM type [1,2,3]. 

The disease involves multiple biochemical alterations that are manifested by changes in the chorioretinal neurovascular couple (NVC) and its cellular components [4,5], causing a series of morphological and functional manifestations in the ocular fundus as well as acute or progressive vision impairment and blindness, mainly mediated by the appearance of vitreous hemorrhage and/or (DME) [6,7]. A major risk factor for DR is chronic hyperglycemia. Moreover, DR duration, obesity, hypertension blood pressure (HBP), dyslipidemia, and other ocular and systemic diseases have been considered in the onset and progression of retinopathy [6,7,8]. Population studies have also described DR as more likely to appear in diabetics with a family history of the disease. Major non-modifiable risk factors for DR are age, race, and genetic profile. It is worth mentioning that enhancing metabolic DM control results in a significant reduction in eye complications [8,9,10,11].

The development of clinical DR severity scales is based on the scientific evidence for covering the real needs for solid clinical classification protocols for DR and DME. In fact, the five-stage DR severity classification includes three stages of low–mild risk, severe non-proliferative retinopathy (NPDR), and proliferative retinopathy (PDR). Diabetic macular edema (DME) is classified as apparently present or absent. Therefore, PDR is a long-term complication of diabetes [8,9,10,11,12]. In this stage, there are visible microaneurysms, intraretinal hemorrhages, retinal edema, waxy lipid exudates, venous dilations, intraretinal microvascular abnormalities, cotton wool exudates, arteriolar abnormalities, and areas of capillary closure. Moreover, the most important process occurring in this stage is neovascularization caused by abnormal angiogenesis in addition to the inflammation and expansion of extracellular matrix (ECM) components, resulting in the outgrowth of fibrovascular membranes at the vitreoretinal interface, leading to severe complications such as vitreous hemorrhage and tractional retinal detachment [7,8,9,10,11,12].

There is presently no cure for DR. Current diagnostic and therapeutic tools and outstanding new perspectives, including the digital revolution for health care (telemedicine, artificial intelligence, wearable sensors, new apps, etc.) [13,14], and enabling innovative medical research in risk factors, pathophysiological mechanisms, and underlying predictors [15,16,17,18,19,20,21] may help to significantly gain insights for diabetic eye care and management.

Currently, the most widely used key actions for DR/DME include strong metabolic control and the occurrence of related comorbidities (hypertension blood pressure (HBP), dyslipidemia, obesity, etc.) [15,22,23,24]. However, for the advanced cases, laser treatment, intravitreal therapy (anti-angiogenic, steroid), and vitreoretinal surgery are required. Concerning the prevention of progression, it is widely accepted that panretinal laser photocoagulation is useful for retinal ischemic disease, and grid, sub-threshold diode micropulse laser photocoagulation is useful for DME [25,26,27]. Furthermore, intravitreal injection of anti-angiogenic and/or steroid treatment have been extensively used for PDR and DME [25,26,27,28,29]. Vitrectomy is conventionally performed in cases of vitreous hemorrhage and/or tractional retinal detachment [29]. Despite the existence of the described DR/DME treatments, a significant number of patients experience complications and visual impairment. Thus far, no curative treatments for DR exist, and there is still a long road to effectively avoiding visual impairment and blindness in diabetics.

Reactive oxygen species (ROS) damage nucleic acids, lipids, and proteins [30,31,32]. ROS can modify cell signaling proteins, with functional consequences involving pathological processes. In fact, excessive ROS function as signaling amplifiers to specifically activate cellular stress pathways (protein kinase C (PKC), mitogen-activated protein kinase (MAPK), and nuclear factor kappa B (NF-κB)), leading to the activation of inflammation, angiogenesis, and apoptosis [22,30,31,32]. Polyol pathway activation (aldose reductase converting excess glucose to sorbitol) contributes to oxidative stress (OS) by increasing the NADH/NAD+ ratio [23,33]. Advanced glycation end-products (AGEs), formed by the process of nonenzymatic binding of glucose to proteins, lipids, and nucleic acids, can lead to important changes in the structure and function of proteins, the generation of OS, and inflammation, with the increased expression of lipid peroxidation (LPO) byproducts [30,31,32,33,34,35,36,37,38], pro-inflammatory cytokines [39,40], and growth factors [37,41,42].

Moreover, there is growing evidence that neurodegeneration may also be an independent process in DR [4,5,36,37,38,39]. Regarding its pathophysiology, increasing research has demonstrated that DR is a complex neurovascular disease in which chronic hyperglycemia affects both the retinal vasculature and neural tissue [39]. The neurovascular couple (NVC) concept was introduced from the CNS to the retina at the beginning of the millennium [4,5] by referring to the interrelationship between neuronal, glial, immune, and vascular retinal cell phenotypes, that became unable to maintain homeostasis and inner BBR integrity, as well as to appropriately orchestrate blood flow responses to the high metabolic retinal demand. Epidemiological and experimental studies have reported that blood flow regulation and NVC deficiencies may exist before the presence of neural dysfunction in the diabetic retina [39]. In addition, the accumulation of AGEs and the induction of PKC, polyols, and hexosamine pathways have been classically considered pathogenic mechanisms of DR [35,36]. In this regard, OS, neuroinflammation, and glutamate-dependent excitotoxicity result in decreased retinal cell survival. Finally, the alteration of the NVC in the retina leads to hypoxia and progressive neurodegeneration, neuroinflammation, gliosis, BRB dysfunction, edema, angiogenesis, and fibrosis with increasing expression of several molecules, such as cytokines and pro-angiogenic factors (as the VEGF), with all of the abovementioned processes being interconnected during the DR/DME course [4,5,22,30,31,32,33,34,35,36,37,38,39,40,41,42,43,44,45].

In our previous research, the expression levels of pro-oxidant (malondialdehyde: MDA; 4-hydroxynonenal: 4HNE), pro-inflammatory (interleukins: (IL) IL1β, IL6; tumor necrosis factor (TNF)-α), pro-angiogenic (vascular endothelial growth factor: VEGF), and pro-apoptotic [caspase 3: CAS3; poly adenyl ribose polymerase (PARP) 1] proteins increased to hazardous levels, whereas the levels of antioxidants (superoxide dismutase: SOD; catalase: CAT; total antioxidant capacity: TAC) decreased in DM2 patients with NPDR and PDR [41,42,43,44,46,47]. The abovementioned results and increasing experimental and clinical evidence strongly specify that OS, inflammation, angiogenesis, and apoptosis are involved in the pathogenesis of PDR/DME, independently or all together, responding to a functional link in the diabetic retina [35,36,37,38,39,40,41,42,43,44,45,46,47]. Taken together, these findings indicate that structural/functional damage to the vascular endothelial cells, pericytes, and resident glia induces deleterious changes in the retinal neurons and ultimately their apoptotic death, representing pathogenetic hallmarks in DR/DME [30,31,32,33,34,48,49,50,51,52,53]. 

Given the relatively recent trend toward biomarker discovery and the translation to clinical practice, we aimed to explore the cascade of events related to oxidative stress, inflammation, angiogenesis, and apoptosis pathological pathways and the correlation of data obtained from blood and vitreous humor samples from PDR and DME patients and surrogate controls.

Therefore, the present study aimed to extend current knowledge on the biochemical and molecular mechanisms involved in DR and DME by exploring the cascade of events related to the oxidative stress, inflammation, angiogenesis, and apoptosis pathological pathways, in addition to the correlation of data obtained from blood and vitreous humor samples from DM2 patients, thereby validating previous results as well as addressing potential issues for precision medicine in DR/DME diagnostic and therapy, based on hallmark mechanisms of actions. The identification of more specific and sensitive biomarkers is relevant for facilitating the earlier detection of DR/DME toward reducing the prevalence of visual disability.

According to our viewpoint and based on the above statements, unraveling the manner in which the molecular mechanisms of the above signaling pathways participate in DR and DME is essential for ensuring better eye and vision care in diabetics. 

## 2. Results

At baseline, 205 volunteers were recruited; however, only 185 suitable participants were included in the study. After the ophthalmological examination, vitreoretinal surgery, and sampling procedures, 160 participants were definitively included for statistical processing and distributed into the following: T2DM patients with PDR (n = 64) and/or DME (n = 48) and surrogate controls (n = 48) (see the flowchart with the recruitment characteristics and proceedings of this study in the Material and Methods section). Constituting the SCG of non-diabetic patients were non-complicated macular hole (n = 12), epiretinal membranes (n = 15), and rhegmatogenous retinal detachment (n = 21) participants.

Combined patient and sampling factors led to a mild reduction (11%) in the initial number of participants, and their corresponding data were excluded from statistical processing.

The remainder of this section is subdivided to facilitate better interpretation of the study results.

### 2.1. Sociodemographic and Patient Characteristics

The mean age of the participants was 60 ± 10 years. Distribution by age and sex and other participant characteristics are listed in Table 1. The study groups enrolled in this study (64 in PDRG; 48 in DMEG; 48 in the SCG) showed no statistically significant differences in age and DM duration.

### 2.2. Ophthalmologic Examination

A systematized ophthalmological examination was performed on all participants, and the main data are shown in Table 2. Among all DR patients (n = 112), 64 were classified as PDR cases and 48 had evidence of DME on OCT. BCVA in each eye was significantly lower in PDRG and DMEG compared with SCG (*p* < 0.001). IOP in each eye was significantly higher in DMEG than in the CG (<0.001). The CSFT was significantly higher in PDRG vs. the CG (*p* < 0.001). In fact, both parameters, CSFT and CAT, increased with DR severity according to the macular cube 512 × 128 protocol. Moreover, when comparing the three study groups (64 in PDRG; 48 in DMEG; 48 in SCG), statistically significant increases in CSFT and CAT, as well as reduction in BCVA was detected (0.001; 0.0001; 0.0001).

### 2.3. Bioanalytical Testing

#### 2.3.1. The Blood Parameters of the Study Groups Are Shown in Table 3 and Figure 1, Figure 2 and Figure 3

The glycemic profile was significantly higher in the T2DM groups with respect to the SGC. In addition, the total cholesterol C reactive protein levels in the blood samples were significantly higher in diabetics versus the CG.

**Table 3 ijms-24-08227-t003:** Biochemical parameters of the study participants.

	SCG	PDRG	DMEG
Glucose (mg/dL)	83.47 ± 7.95	138.71 ± 42.45 *	184.35 ± 42.11 *
HbA1c (%)	5.48 ± 0.36	7.04 ± 0.76	7.76 ± 0.96
Total Cholesterol (mg/dL)	158.73 ± 20.21	220.35 ± 40.41	235.51 ± 39.44 *
C-Reactive Protein (mg/dL)	1.73 ± 7.95	3.24 ± 0.71 *	4.08 ± 0.79 *

SCG: surrogate control group; PDRG: proliferative diabetic retinopathy group; DMEG: diabetic macular edema group; HbA1c: glycated hemoglobin. Values are the mean ± SD. * *p* < 0.01.

**Figure 1 ijms-24-08227-f001:**
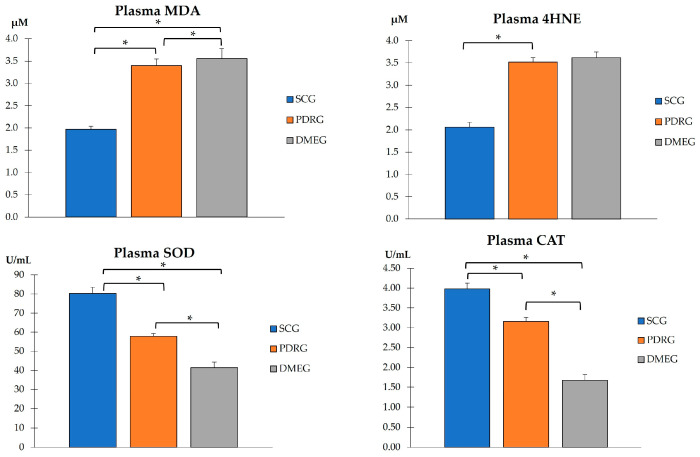
PLS pro-oxidants (MDA and 4HNE), and antioxidant enzymes (SOD and CAT) in the study groups. Values are mean ± SD. * *p* < 0.001. PLS: plasma; MDA: malondialdehyde; 4HNE: 4-hydroxynonenal; SOD: superoxide dismutase; CAT: catalase. SCG: surrogate control group; PDRG: proliferative diabetic retinopathy group; DME: diabetic macular edema group.

**Figure 2 ijms-24-08227-f002:**
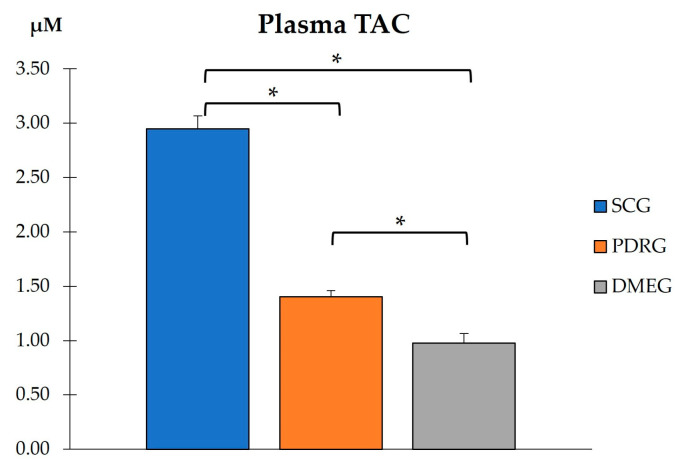
PLS total antioxidant capacity in the study groups. Values are mean ± SD. * *p* < 0.001. PLS: plasma; TAC: total antioxidant capacity; CG: surrogate control group; PDRG: proliferative diabetic retinopathy group; DMEG: diabetic macular edema group.

**Figure 3 ijms-24-08227-f003:**
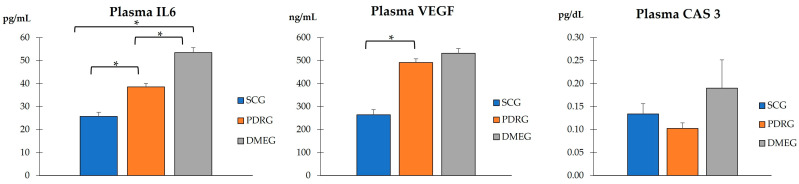
PLS concentration of molecules involved in inflammation, angiogenesis, and apoptosis in the study participants. Values are mean ± SD. * *p* < 0.001. PLS: plasma; IL6: Interleukin 6; VEGF: vascular endothelial growth factor; CAS3: caspase 3; PDRG: proliferative diabetic retinopathy group; DMEG: diabetic macular edema group; SCG: surrogate control group.

#### 2.3.2. Plasma Parameters of the Study Groups

The assayed parameters in the PLS samples of the study participants are reflected in the following results.

When comparing the PLS-TAC from PDRG and DMEG with the results from the SCG, our data show statistically lower values in both diabetic study groups (Figure 2). The data show that the antioxidant defenses failed in counteracting pro-oxidant formation in diabetics.

Figure 3 shows the PLS expression levels of the pro-inflammatory IL6, pro-angiogenic VEGF, and pro-apoptotic CAS3 proteins of PDRG and DMEG compared with those of SCG. Significantly higher levels of IL6 and VEGF were detected in the PLS of diabetics than in the non-diabetic participants. A comparison of the CAS3 PLS levels among groups did not indicate significant differences.

#### 2.3.3. Vitreous Body Parameters

The assayed parameters of the VIT samples are reflected in the following illustrations.

Both LPO parameters in the VIT-MDA/4HNE showed statistically significant higher values in PDR and DME patients than in the participants in the SCG (Figure 4). Furthermore, a comparison of the VIT-TAC showed significantly lower values in the diabetic groups than in the SCG (Figure 4). Our results suggest that the antioxidant defenses were ineffective in counteracting the pro-oxidant dominancy in the VIT from diabetics, being the most relevant change observed in DME patients.

The VIT-TAC values were significantly lower in the samples of the diabetic groups than in those of the SCG (as reflected in Figure 5). The DMEG had the lowest VIT-TAC values among the three groups of participants. Moreover, the VIT expression levels of the pro-inflammatory IL6, pro-angiogenic VEGF, and pro-apoptotic CAS3 proteins are shown in Figure 6. Significantly higher levels of IL6 and VEGF were detected in the VIT of PDRG and DMEG compared with that of the SCG. A comparison between groups showed a lack of significant differences in the VIT-CAS3 levels.

In summary, we prepared a new table, reflecting the biomarkers obtained from the plasma and vitreous body samples, of the data comparison of PDR and DME patients. Table 4 includes the assayed molecules and their corresponding units of measurement. The significance level is also specifically shown for each variable, being the most noticeable differences between groups: VIT-MDA (*p* = 1.06 × 10^−22^) for PDRG vs. SCG, VIT-4HNE (*p* = 4.44 × 10^−16^) for DMEG vs. SCG; PLS-TAC (*p* = 9.75 × 10^−14^) and VIT 4HNE (*p* = 8.96 × 10^−14^) for PDRG vs. SCG).

#### 2.3.4. Correlation Analysis

We also conducted correlation analyses of PDRG and DMEG of our study participants based on clinical, biochemical, and molecular data, aimed toward improving knowledge on the ophthalmologic variables, glycemic profile, and molecular fingerprint, which may help rapidly identify T2DM patients at high risk of developing PDR and DME and to better manage and prevent visual impairment and blindness in diabetics.

The statistical processing (Figure 7) showed several significant correlations between the analyzed variables. The vast majority of the PLS and VIT levels of the main LPO by-products (MDA, HNE) and the antioxidant parameters (SOD, CAT, TAC) showed noticeable differences between groups, but lacked significant correlation between them, neither in PDRG or DMEG. However, in the PDR group, we can highlight a significant positive correlation between the PLS-CAT and the VIT-SOD levels (rho = 0.5; *p* < 0.01). In DMEG, the PLS-4HNE levels showed a negative correlation with the VIT-TAC levels (rho = −0.5; *p* > 0.01).

When comparing the ophthalmological variables with the same pro-oxidant and antioxidant parameters (MDA; HNE, SOD, CAT, TAC) in the PDR group, we saw a significant positive correlation between CSFT in the LE and the VIT-SOD levels (rho = 0.5; *p* < 0.01), and a significant negative correlation between CSFT in the RE and the VIT-4HNE, as well as the IOP in the LE and the VIT-SOD (rho = −0.5; *p* < 0.01). In the DME group the analysis showed a negative correlation between CSFT in the RE and the VIT-CAT, the IOP in the LE and VIT-MDA, and the BCVA in the LE and the VIT-4HNE (rho = −0.5; *p* < 0.01).

## 3. Discussion

In this work, we intended to address potential issues for precision medicine in PDR/DME diagnostics and therapy based on the hallmark mechanisms of actions. The identification of more specific and sensitive biomarkers that have a known relationship with a specific clinical endpoint is relevant for facilitating the earlier detection of PDR/DME and reducing the prevalence of visual disability. By using plasma and vitreous body samples, we found significantly higher loads of the LPO byproducts MDA and 4HNE, as well as pro-inflammatory CPR and IL6 and pro-angiogenic VEGF. Significantly lower activity of SOD, CAT, and TAC was observed in the T2DM patients with PDR (after excluding other DM types and/or patients with NPDR) and the DME patients compared with the surrogate controls.

The pathophysiology of DM and DR has been widely investigated. Epidemiological and experimental studies have demonstrated a powerful connection between chronic hyperglycemia and poor metabolic control with the onset and/or progression of DR [3,5,6,7,8,9,10,11,12,15,17,19,20]. The first clinical manifestations of DR are present when BRB rupture occurs and triggers hyperpermeability of the retinal vessels [22,37,54,55,56]. Liu et al. [57] reported the important role of claudins (major transmembrane protein constituents of tight junctions in endothelial and epithelial human tissues) in barrier and pore formation, in addition to their influence in tight junctions. Similarly, other components critical to the development of DR involve OS and antioxidant status in the PLS and VIT of PDR/DME patients [30,31,32,33,34,35,36,37,38,48,52,53,54,55,56,57,58,59,60,61,62]. Xia et al. [63] recently reviewed the intravitreal molecules involved in DR, the increasing role of oxidative stress (lipid peroxides, SOD), inflammation-related actors (IL1β, IL6, IL13, IL37, interferon-γ), angiogenesis (VEGF), and the kallikrein system (bradykinin, PLS kallikrein, coagulation factor XII), as well as the introduction of new factors such as adiponectin, non-coding RNAs, and renin (pro) receptor. In fact, increased ROS formation leads to a significant repression of mitochondrial biogenesis, inducing decreased antioxidant defenses (SOD), resulting in mitochondrial dysfunction and thus altering the mitochondrial outer membrane, which in turn provokes the release of cytochrome C, leading to apoptotic cell death [32,38,45]. In addition, excessive ROS can trigger the immune response with the release of inflammatory factors (NF-κB), PKC, MAPK, etc., with the intensification and chronification of the inflammatory process [63,64,65,66]. Antioxidant defenses include enzymatic systems, with SOD being the main chain-breaking antioxidant, acting in the aqueous phase by trapping superoxide anions via the dismutation reaction [58]. In the present work, significant differences were detected in SOD activity in the PVR and DME groups in comparison with the surrogate controls. It has to be said that we did not discriminate between the Cu-Zn and Mn SOD isoforms, but an important reduction was observed in the total SOD activity in the VIT samples from PDRG compared with SCG, which is in accordance with previous related reports [58,60]. CAT is an antioxidant enzyme which plays a pivotal role in protecting cells and tissues against the harmful effects of hydrogen peroxide [59,60]. Moreover, it is important to point out that significantly lower CAT activity in PDR and DME eyes was also demonstrated in a previous report of our research group [48,60], although others have described a significant increase in the CAT levels in diabetics, probably to adaptive mechanisms in response to pro-oxidants in diabetics [33,34,36,61]. It is well established that antioxidant activity undergoes significant age-related reduction in the body [31], which could explain, in part, the differences in the above study results.

The mean age of our study participants was 60 ± 11 years, without significant differences between groups. However, in the comparison between the cases and surrogate controls, a significant reduction in the endogenous antioxidant systems was observed in both groups of diabetics, which is consistent with previous reports [48,49,50,51,52,53,62].

On the basis of the results mentioned above, OS-related molecules have recently been a point of focus for DR/DME researchers, with the conclusion that biomarkers of oxidative stress are significantly higher in the PLS and VIT of PDR patients compared with controls. In fact, the SOD and LPO levels were found to be significantly higher (with a positive correlation between them), whereas the MDA did not reflect significant differences between groups. Conversely, the present work demonstrates that the MDA and the 4HNE were significantly elevated whilst the antioxidant load significantly decreased in the PLS and VIT of PDR/DME patients. Our data suggest that the combination of increased MDA/4HNE and decreased SOD/CAT/TAC can serve as a potential biomarker of DR progression. Moreover, we conducted correlation analysis between the PLS and VIT variables in PDRG/DMEG and a positive correlation between the PLS-CAT and the VIT-SOD levels (rho = 0.5; *p* < 0.01) in PDRG, and a negative correlation between the PSD-4HNE and VIT-TAC levels were detected (rho = 0.5; *p* < 0.01) in DMEG. To summarize, the above molecules can be considered potential biomarkers for discerning the risk of PDR/DME in T2DM patients.

It is well known that cellular interactions in the context of biochemical and metabolic dysregulation (polyol pathway, PKC activation, AGI upregulation, and renin angiotensin system activation) occur in the diabetic retina. Therefore, the described biochemical changes result in the induction of a series of events, such as oxidative stress, inflammation, angiogenesis, and apoptosis, which injury the retina. Cellular damage, loss of vascular endothelial cells and pericytes, and basement membrane thinning are important initiators of DR pathology. Cytokine (IL1β, IL6, TNF-α) increases and loss of neuroprotective and neurotrophic molecules also occur in a hyperglycemic environment [38,39,40,41,45,65,66]. From a molecular viewpoint, persistent hyperglycemia facilitates the formation of ROS and pro-inflammatory mediators. Increased ROS activates inflammatory factors, and in addition, the immune response increases ROS formation in a stepwise fashion, strongly suggesting there is crosstalk between these pathologic pathways [38,42]. Then, hypoxia triggers the expression of pro-angiogenic factors, of which VEGF is the most representative [41,67,68,69].

Hereafter and for practical purposes, we summarize the pathogenesis of DR in three components: (1) vaso-fibro-proliferative; (2) neuroinflammatory; and (3) neurovascular.

First, the vaso-fibro-proliferative component can be explained through ischemia–reperfusion mechanisms, leading to an appropriate environment for ROS generation and the activation of signaling pathways, such as the angiogenic, inflammatory, and apoptotic routes [33,34,35,36,37,65,66,67,68,69]. Chronic hyperglycemia induces high levels of VEGF, a core molecule that disrupts the balance between pro-/anti-angiogenic factors, favoring the pro-angiogenic atmosphere [41,45]. As a consequence of this, VEGF becomes engaged in destructive action against vascular endothelial cells, stimulating the generation of angiogenic neovessels, which marks a milestone of disease progression to more severe stages, PDR [67,68,69].

In the PLS and VIT samples of our study participants, we detected a differential expression profile of VEGF between PDRG/DMEG and CG (see Figure 3 and Figure 6). In fact, other researchers also reported a significant increase in VEGF expression levels in the PLS and VIT samples of PDR/DME patients compared with the controls [67,68,69]. Indeed, Wang et al. [70] studied 50 patients with PDR who underwent pars plana vitrectomy and 56 healthy controls. The VEGF levels were assayed in PLS and VIT samples by enzyme-linked immunosorbent assays (ELISA), and the data showed significantly higher values in both samples of the PDR patients, and, interestingly, the VIT VEGF concentrations were positively associated with the PLS VEGF levels in these patients. The abovementioned results are in agreement with ours, and we found a correlation of the VEGF expression in samples of both the PDR/DME study groups (see Table 4 and Figure 7).

Next, we comment on another critical component of PDR/DME: inflammation. Chronic hyperglycemia induces inflammation and polynuclear neutrophil activation. Cytokines/chemokines are also implicated in the mechanisms of modulating the immune response, acting as chemotactic stimuli for inflammatory cells and fibroblasts. The resident immune cells change their phenotype, becoming the primary source of a variety of molecules, including collagen [71]. Because of this, a more conducive environment has been created for the formation of membranes and tractional bands between the retina and vitreous, commonly leading to retinal tears and detachments. Hence, it is worth mentioning the elevated levels of CRP and IL6 in PLS samples, as well as of IL6 in VIT samples, of our diabetic participants (see Table 3 and Figure 3 and Figure 5). As published in other reports, the expression levels of cytokines have been correlated with the progression of retinopathy to advanced stages and the appearance of DME, the leading cause of vision loss in diabetics [26,27,65,71], which are supported in our work. A hydroelectrolytic imbalance between the intracellular and extracellular space in the diabetic retina results in increased water diffusion and inner BRB breakdown [44,63,70]. In this process, the altered metabolism of neuroglial cells has a leading role [66]. Subsequent cellular swelling and the extravasation of liquid to the extracellular space (which mainly accumulates in the external plexiform membrane and the inner and outer nuclear retinal layers), results in the appearance of cyst spaces [72,73,74]. To sum up, the cascade of processes that we present above induces retinal hypoxia with BRB breakdown, the increased expression of VEGF, activation of the immune response, and the appearance of DME [70,72,73,74]. As a result of the correlation analyses of the present work, considering biochemical and molecular factors in the T2DM patients, we demonstrated that PLS 4HNE, VEGF, and IL6 and VIT TAC were the most significant molecules for DME. We speculate that those molecules are the best predictors for DME in our diabetic study population.

Recently, the concept of DR as a neurovascular disease has gained many followers. It is well recognized that neurons, neuroglia, and vascular cells intimately interact with each other to maintain a healthy retinal environment, maintaining their well-being, and to perform all physiological functions. In fact, DR is a complex neurovascular disorder in which sustained hyperglycemia influences the retinal vasculature and neural tissue [39]. Accordingly, the NVC approach was introduced from the CNS to the retina in the early years of the 21st century [4,5]. With respect to the NVC, the dysfunctional interrelationship between neurons, glial, immune, and vascular retinal cell phenotypes results in a failure to maintain homeostasis and inner BBR integrity and to harmonize blood flow responses to the high metabolic retinal demand [44]. In the PLS and VIT samples of our study participants, we assayed molecules involved in OS, inflammation, angiogenesis, and apoptosis to improve our knowledge of the physiopathology of PDR/DME. Our results regarding CAS3 are unexpected and require further attention. In fact, CAS3 is a cysteine protease with aspartic acid specificity involved in apoptosis. It has been reported that CAS3-targeted isatin analogs could be imaged (by positron emission tomography) in addition to the activation of CAS3 in response to anticancer therapy based on the induction of apoptosis [75]. Furthermore, CAS3 has been linked to glaucoma neurodegeneration because of its role in apoptosis [76]. No significant differences in the CAS3 expression levels of the PLS and VIT samples were detected between PDRG and DMEG compared with SCG. However, Tian et al. [77] studied 20 NPDR patients, 20 PDR patients, and 20 patients with idiopathic macular hole as the control group that underwent vitrectomy, to collect vitreous samples. Venous blood was also collected. The concentration of CAS3, among other molecules, was assayed by ELISA. The authors concluded that DR severity is positively correlated with CAS3. Taking into consideration that apoptosis is an important step in the pathogenesis of advanced DR, further research is needed to understand our CAS3 results in the PDR and DME patients.

There have also been many reports on the correlation between DR and local and systemic risk factors and biochemical and molecular genetics data, and the results of these research studies are promising [3,6,12,15,17,19,20,78,79]. However, due to the heterogeneity of the studies (ethnic, age range, gender, work design, molecular assays, etc.), the assessment of PDR and DME risk in T2DM patients is biased and not easily transferrable to the global use of specific biomarkers in diabetics. Because of this, the key objective of the present study was to identify clinical, biochemical, and molecular biomarkers to reduce the risk of worsening the DR/DME course and diabetic blindness.

We first presented physiopathological knowledge of DR/DME, including the risk factors that cause retinopathy and the clinical and imaging indicators of disease progression. Then, we showed data from the classic biochemical parameters. Finally, we presented the results from the molecules involved in the most relevant pathogenic mechanisms, such as oxidative stress, inflammation, angiogenesis, and apoptosis in addition to the differences between groups. We also conducted correlation analysis between PDRG and DMEG on the basis of specific controlled variables.

The strengths of this study include the homogeneous population-based sampling. This work provides an opportunity to identify new biomarkers for the preclinical diagnosis and better therapeutic management of PDR/DME. With objective assessment of PDR/DME, the diagnostic error can be reduced, and the precision of estimates can be improved. This work will be continued by increasing the power to detect associations with risk factors, comorbidities, and PDR/DME therapy. All of these aspects may help to improve knowledge and to solve uncertainties that may emerge in the course of the research.

The study limitations have been considered and are as follows: Participants may have provided inaccurate answers to our questions on the issues concerning the characteristics of the disease, comorbidities, and treatments. We did not enclose the treatments for the eyes and/or systemic diseases. With the study database and statistical processing, a large amount of information was set up. In this context, we mainly focused on the study objectives. Because of this, some of the information and data have been omitted from the final data in processing. In trying to diminish the above study limitations, some specific actions were taken, among them the revision of the patient’s clinical charts with the patient themself and an accompanying person. We also conversed among ourselves regarding any discrepancies arising from the selection of suitable participants, data screening, and the results. To ensure maximum coherence in the registered information, data scrubbing and normalization were independently performed by two researchers. Moreover, we solicited special help from an experienced retina specialist for better estimation of contradictions. With the above interventions, we intended to improve the power of our data.

In summary, key research in PDR/DME focuses on the molecular genetic mechanisms (biomarkers, gene editing), advanced diagnostic techniques (deep learning automated algorithms for imaging, robotics), and application of stem cell appliances. This work aimed to revisit current knowledge and to recognize the most pressing problems in DR/DME, such as preventing and treating DR/DME, which are major concerning challenges in this field. New strategies are urgently needed for preclinically detecting potential vision-threatening retinopathy and maculopathy and prevent their progression in diabetics.

We demonstrated that the MDA, 4HNE, VEGF, and IL6 concentrations increased whilst SOD, CAT, and TAC decreased in the PLS and VIT samples from PDR and DME patients versus the surrogate controls. We observed a positive correlation between the PLS-CAT and the VIT-SOD levels in PDRG, and a negative correlation between the PSD-4HNE and the VIT-TAC levels in DMEG. When the retinal imaging variables were integrated, data showed a positive correlation between the CSFT and the VIT-SOD levels, and a negative correlation between the CSFT and the VIT-4HNE levels in PDRG. In DMEG, the CSFT displayed a negative correlation with VIT-CAT.

We suggest that OS, angiogenesis, and inflammation play important roles in the pathogenesis of PDR and DME, with the altered immune response critically contributing to the development and progression of macular edema.

As the formation of ROS and their harmful effects on biomolecules can be reduced/avoided by blockage and pro-oxidants removed by means of antioxidant enzymes and scavengers, further research on a possible coadjunctive therapeutic approach to PVR and DME is needed. This will allow important adjuvant intervention in the current anti-inflammatory and anti-angiogenic therapy of the diabetic retina, aiming to counteract visual impairment and blindness in the affected patients.

## 4. Materials and Methods

### 4.1. Study Design

The present study adhered to the tenets of the Declaration of Helsinki (Edinburgh, 2000). We met the requirements for clinical research and had the approval of the Ethics Committee of the study centers (42.22/6 May 2022) as well as maintained the data privacy of the study participants. Sample size was estimated using the epicalc package included in the shareware R program (https://www.statmethods.net/stats/power.html, accessed on 15 January 2023). Therefore, the study was powered to detect statistical differences between groups. The sample size was larger than that of other similar publications, and other researchers worked with a larger number of participants than ours. A prospective, observational, multicenter case–control study was conducted between 2021 and 2022. The sampling methods were previously described [31,32,33,34,35,36,48,49,50]. The main objective of this study was to outline the T2DM patient characteristics, risk factors, specific blood and vitreous body-based biochemical parameters and precise DR hallmarks to achieve the differential fingerprint of potential clinical and molecular biomarkers for DR that may be used for the better management of the health and vision care of T2DM patients with PDR/DME. The independent pseudo anonymized data and a safety monitoring committee provided oversight for the study.

### 4.2. Study Participants

A total of 205 participants of both sexes, aged 40–80 years, were pre-selected by ophthalmologists from the retina sections at four collaborative hospitals located in Spain (Clinic of Barcelona, Clinic of Valladolid, Punta Europa of Algeciras, and Dr. Peset of Valencia) according to the study inclusion/exclusion criteria (listed in Table 5).

The 205 initial potential participants were interviewed according to the study purpose and appointed to the first selection step.

Sociodemographics, personal characteristics, treatment, familial history, and comorbidity data were recorded at the initial visit. In addition, the participants underwent ophthalmological examination of both eyes. Suitable participants were classified as (1) 112 T2DM patients and (2) 48 nondiabetic individuals that were programmed for vitrectomy as the surrogative control group (SCG).

Finally, a total of 160 volunteers (112 T2DM with PDR/DME and 48 from the CG) were eligible for the study, verbally informed by the ophthalmologist, and signed the consent form. The screening procedures for the recruitment and appointment schedules of the study participants are reflected in the flowchart (Figure 8).

### 4.3. Ophthalmic Examination

The T2DM participants were those who presented a PDR/DME diagnosis (as shown in Table 5) as defined by the International Council of Ophthalmology (ICO) severity scale report [1,80]. Indications for vitrectomy in these participants included a best-corrected visual acuity (BCVA) value lower than 20/40 due to vitreous hemorrhage of at least 2 months duration.

The CG participants were those who presented idiopathic MH, ERM, or RRD and no DM, which were evaluated to rule out history or signs of other associated eye disease or relevant comorbidity, as reflected in Table 4. MH is caused by vitreomacular traction without any associated retinal ischemia, vascular proliferation, or inflammation (this condition is least likely to be associated with the local release of VEGF or inflammatory cytokines) [81]. The ERM, also named cellophane maculopathy or macular pucker, is a thin sheet of a fibrocellular, avascular, semitranslucent membrane that forms on the inner retinal surface [82]. The precise diagnosis of MH and ERM was achieved by means of OCT macular examination. RRD diagnosis was based on ocular fundus and spectral domain (SD) optical coherence tomography (OCT) examination [83], and cases with macular involvement were excluded from this study. SD-OCT was also an essential tool for addressing the feasibility and prognosis of vitreoretinal surgery in each condition.

The RE and LE from each participant were examined, and different measurements were conducted: BCVA, the logarithm of the minimum angle of resolution (LogMAR), slit-lamp biomicroscopy (ImageNet; Topcon, Barcelona, Spain), intraocular pressure (IOP) by Goldmann applanation tonometry (Haag-Streit AT 900; Köniz, Switzerland), color fundus photography (CFP) (ImageNet; Topcon, Barcelona, Spain), and SD-OCT (Carl Zeiss Meditec, Madrid, Spain). All data were recorded. For this, OCT image analysis of the retina was conducted, and the OCT parameters (CSFT: central subfield foveal thickness; CAT: cube average thickness) were explored using a direct cross-sectional imaging device.

Retinal imaging and data records were evaluated by retina specialists to obtain optimal PDR diagnosis grading in each eye according to the severity scale of DR and DME for the ICO [1,3,8]. As globally considered, the presence and number of microaneurysms/hemorrhages, venous beading, and intraretinal microvascular abnormalities were taken for DR diagnosis. Worsening was regarded when the above signs augmented during the study course. Focal or diffuse DME was also taken into consideration according to the ICO staging, as follows: (1) no EMD, (2) EMD without central compromise (macular thickening without central subfield affectation (1 mm diameter), and (3) EMD with central subfield compromise. The color fundus photographs and OCT parameters were evaluated by four independent observers at the collaborative hospitals in Spain (J.C-E., S. di L., R.C.M, V.A.I.). In addition, an international experienced retina specialist from the John Hopkins Hospital in USA (J.F.A.) was solicited for the estimation of contradictions.

### 4.4. Sampling Procedures

Two different types of biological samples were extracted from the study participants: venous peripheral blood and vitreous body samples.

#### 4.4.1. Blood Sampling

Blood sampling was scheduled under fasting conditions and collected, stored, and processed to conduct biochemical tests. In this sense, antecubital vein blood was collected into 4.5 mL ethylenediaminetetraacetic acid (EDTA) or sodium citrate vacutainer tubes (Becton Dickinson, Auckland, New Zealand), as anticoagulant, at 8:00 a.m. of the scheduled day. One EDTA tube (purple cap) from each patient was delivered to the Department of Clinical Analysis of the University Hospital Dr. Peset (Valencia, Spain), where it was used to determine the concentrations of glucose, glycosylated hemoglobin (HbA1c), total cholesterol (Chol), and the inflammatory markers C-reactive protein (CRP) and interleukin (IL)6. The citrate-containing tube was centrifuged at 3000 rpm/10 min to obtain the PLS fraction, which was aliquoted and stored at −80 °C until processing at the laboratories of the Ophthalmic Research Unit “Santiago Grisolía” (Valencia, Spain) and laboratories of the Surgery Department at the Faculty of Medicine and Odontology of the University of Valencia (Valencia, Spain). These PLS samples were used to determine the concentrations of malondialdehyde (MDA), 4-hydroxynonenal (4HNE), superoxide dismutase (SOD), and catalase (CAT) and to quantify the total antioxidant capacity (TAC).

#### 4.4.2. Vitreous Body Sampling

Small samples of undiluted VIT were obtained before isotonic infusion and aspiration of the programmed pars plana microincision vitrectomy surgery. The surgeon performed manual gentle suction via a 2 mL syringe to aspirate the vitreous sample, as previously described [30,31,48,49]. The assistant took the sample (1.0–1.5 mL), which was immediately deposited into labeled test cryotubes and storage boxes to be frozen at −80 °C. At the time of transport to the receiving research center, the sample boxes were transferred via an optimal freezing system during the travel. At reception, the samples were examined, registered, and stored at −80 °C until processing. For each sample, the levels of MDA, 4HNE, SOD and CAT activities, TAC, CAS3, IL6, and VEGF were quantified.

### 4.5. Analytical Laboratory Procedures

Each sample type (total blood, PLS, and VIT) was used to obtain the concentrations of different molecules. All samples were assayed in duplicate. Values were expressed as the mean ± standard deviation, as previously reported [8,26,32,33].

#### 4.5.1. Total Blood Samples

The determination of the glycemic profile (fasting glucose and HbA1c), total cholesterol, CRP, and IL6 were measured by two different automated chemistry analyzers at the Department of Clinical Analysis of the main study center, as follows: (1) Abbott kits manufactured for use with the Architect c8000 (Abbott Laboratories; Abbott Park, IL, USA) and (2) Arkray AU 4050 (Arkray Global Business Inc., Kyoto, Japan). Analytical processing was supervised by a medical specialist.

#### 4.5.2. Plasma Samples

Several molecules were quantified using PLS samples:(1)Lipid peroxidation (LPO) byproducts:
MDA was quantified using the colorimetric TBARS Assay kit (Ref: 10009055, Cayman Chemical Company, Ann Arbor, MI, USA) a thiobarbituric acid (TBA)-based assessment. In the presence of this acidic reactive, MDA forms what is known as TBA reactive substances (TBARS), the amount of which can be quantified by colorimetric methods. The assay was conducted on 100 µL of plasma following the protocol provided by the manufacturer, with the use of a boiling water bath to reach the required temperature (90–100 °C). The reaction product was measured using a spectrophotometer with a light wavelength of 525 nm. The concentration was calculated by extrapolating all standard curve data, as published elsewhere [30,31,33,61].The 4HNE concentration was analyzed using the BIOXYTECH^®^ LPO-586™ Colorimetric Assay for Lipid Peroxidation Markers (Ref: 21012, OXIS Health Products, Inc. Portland, OR; USA). The assay, based on the reaction of a chromogenic reagent, N-methyl-2-phenylindole (R1), with 4-hydroxyalkenals, was conducted following the protocol provided by the manufacturer, using 140 µL of plasma. The reaction occurred under 45 °C of temperature, and the formed product was measured using a spectrophotometer with a light wavelength of 586 nm. The final levels were calculated by extrapolating the standard curve data, as reported before. Since this kit quantifies the concentration of both MDA and 4HNE, the MDA value obtained from the previous kit was subtracted from the value obtained with this kit [61].(2)Antioxidant molecules:
SOD activity was measured according to the techniques described in previous works, based on the ability of SOD to inhibit a superoxide-driven reaction in the presence of EDTA, Mn, Cl, and mercaptoethanol [61].CAT activity was determined using the absorbance technique per unit of time and is a measure of CAT described by analyzing differences between groups using the SPSS/decrease in absorbance at 240 nm [59].The TAC, which is a measure of the combined activities of all of the antioxidants in a sample including vitamins, proteins, lipids, glutathione, and uric acid, was measured in the plasma samples using the colorimetric Antioxidant Assay Kit (Ref: 709001, Cayman Chemical Company, Ann Arbor, MI, USA) based on the antioxidant capacity of the sample to inhibit the 2,2′-azino-di-[3-ethylbenzthiazoline sulphonate] oxidation to 2,2′-azino-di-[3-ethylbenzthiazoline sulphonate] radical solution by the metmyoglobin, as reported. The assay was conducted at room temperature following the protocol provided by the manufacturer, using 10 µL of plasma sample, and the reaction product was measured at a light wavelength of 405 nm using a plate reader. The concentration was calculated by extrapolating all standard curve data [30,49,50,69].(3)Pro-inflammatory molecules:
The IL6 expression was calculated in PLS samples by using the Human IL-6 ELISA Kit (Ref: EH2IL6, Invitrogen, Vienna, Austria), an ELISA-based assay. The assay was conducted at room temperature following the protocol provided by the manufacturer, except that the samples were diluted by 1/2 (25 μL of sample and 25 μL of standard diluent). The reaction product was read twice using a plate reader; first at a light wavelength of 450 nm and then at 550 nm. Then, the 550 nm values were subtracted from the 450 nm values to obtain a corrected value and reduce the interference caused by optical imperfections in the microplate. The concentration of IL6 was calculated by extrapolation of the standard curve data [63].(4)Pro-angiogenic VEGF.
The PLS levels of the VEGF were measured using the Human VEGF ELISA Kit (Ref: KHG0111, Invitrogen, Vienna, Austria), an ELISA-based assay. The assay was conducted at room temperature following the protocol provided by the manufacturer except that the samples were diluted by 1/2 (50 μL of sample and 50 μL of standard diluent). The reaction product was read with a spectrophotometer at a light wavelength of 450 nm, and the final concentration of VEGF was calculated by extrapolation of the standard curve data [41,45,69].(5)Pro-apoptotic CAS3.
The PLS concentration of CAS3 was quantified using the Human Caspase-3 (active) ELISA Kit (Ref: KHO1091, Invitrogen, Vienna, Austria), an enzyme-linked immunosorbent assay (ELISA). The assay was conducted at room temperature following the protocol provided by the manufacturer, using 100 µL of vitreous body sample and reading the reaction product with a plate reader at a light wavelength of 450 nm. The concentration was calculated by extrapolating all standard curve data [50].

#### 4.5.3. Vitreous Body Samples

(1)Pro-oxidants and antioxidants.
The MDA, 4HNE, SOD and CAT levels, and the TAC activity were measured in VIT samples collected from the operating eye of the study participants at the initiation of vitrectomy, stored and processed in the same way as described for the PLS samples [30,31,33,59,61](2)Furthermore, the following molecules were assayed in the VIT samples of the study participants:
Pro-inflammatory IL6. To calculate the concentration of this molecule in the VIT samples, we used the Human IL-6 ELISA Kit (Ref: EH2IL6, Invitrogen, Vienna, Austria), an ELISA-based assay. The assay was conducted at room temperature following the protocol provided by the manufacturer, except that the samples were diluted by 1/2 (25 μL of sample and 25 μL of standard diluent). The reaction product was read twice using a plate reader; first at a light wavelength of 450 nm and then at 550 nm. Then, the 550 nm values were subtracted from the 450 nm values to obtain a corrected value and reduce the interference caused by optical imperfections in the microplate. The concentration of IL6 was calculated by extrapolation of the standard curve data [60].Pro-angiogenic VEGF. The VIT levels of the VEGF were measured using the Human VEGF ELISA Kit (Ref: KHG0111, Invitrogen, Vienna, Austria), an ELISA-based assay. The assay was conducted at room temperature following the protocol provided by the manufacturer except that the samples were diluted by 1/2 (50 μL of sample and 50 μL of standard diluent). The reaction product was read with a spectrophotometer at a light wavelength of 450 nm, and the final concentration of VEGF was calculated by extrapolation of the standard curve data [41,45,69].Pro-apoptotic CAS3. The concentration of CAS3 was quantified using the Human Caspase-3 (active) ELISA Kit (Ref: KHO1091, Invitrogen, Vienna, Austria), an enzyme-linked immunosorbent assay (ELISA). The assay was conducted at room temperature following the protocol provided by the manufacturer, using 100 µL of vitreous body sample and reading the reaction product with a plate reader at a light wavelength of 450 nm. The concentration was calculated by extrapolating all standard curve data [50].

### 4.6. Statistical Processing

All data were recorded in a data sheet of the Microsoft Excel program. The statistical analyses for the data provided in this study were conducted using commercial software of the R Core Team’s Program (R: A language and environment for statistical computing. R Foundation for Statistical Computing, version 4.2.1, Vienna, Austria. ULR: https://www.R-project.org/, accessed on 15 January 2023). Mean ± SD was calculated for each set of data. The Shapiro–Wilk test was conducted to test the normality of continuous variables.

To compare the means between groups, we conducted Student t-tests for normally distributed variables and a Mann–Whitney U test for non-normally distributed variables.

To calculate the correlation between continuous variables, we used the Pearson correlation coefficient for normally distributed variables and the Spearman’s rank correlation coefficient for non-normally distributed variables.

A *p*-value of 0.05 or less was taken as indicating statistical significance for the study goals.

## 5. Conclusions

In this work, the involvement of OS, angiogenesis, and inflammation in PDR and DME was revisited. Our integrated ophthalmologic (ocular examination and fundus imaging) and biochemical–molecular (PLS and VIT samples) data show that (1) chronic hyperglycemia induces anomalies in a variety of biochemical pathways, the majority of these with cross-talk between them; (2) **MDA**, **4HNE**, **TAC**, **VEGF**, and **IL6** are promoters of the clinical manifestations of PDR and DME; and (3) the mechanisms underlying the expression of the above molecules in the diabetic eyes are complex and present challenges.

Therefore, based on our work, the above molecules have been selected as biomarker candidates for distinguishing T2DM patients at risk of PDR, DME and vision loss.

## Figures and Tables

**Figure 4 ijms-24-08227-f004:**
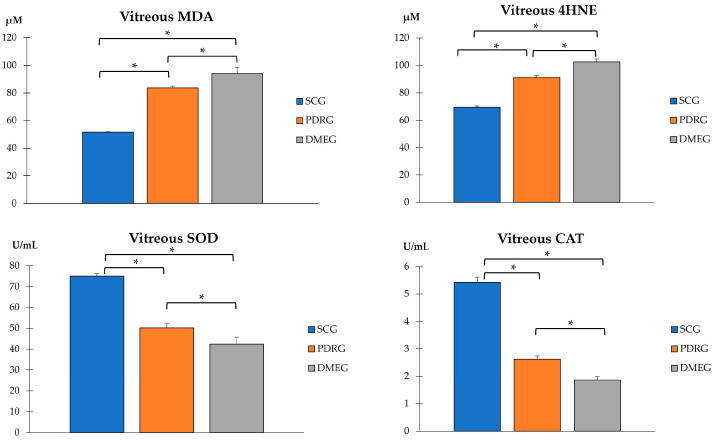
VIT pro-oxidant molecules (MDA and 4HNE), and antioxidant enzymes (SOD and CAT) in the study participants. Values are mean ± SD. * *p* < 0.001. VIT: vitreous; MDA: malondialdehyde (µM); 4HNE: 4-hydroxynonenal; SOD: superoxide dismutase; CAT: catalase. SCG: surrogate control group; PDRG: proliferative diabetic retinopathy group; DME: diabetic macular edema group.

**Figure 5 ijms-24-08227-f005:**
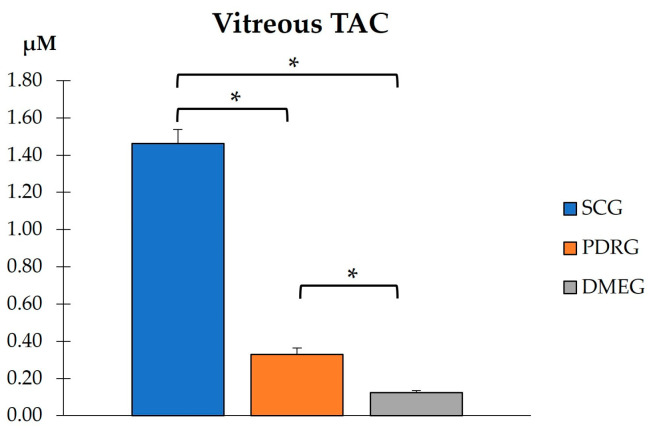
VIT total antioxidant capacity in the study participants. * *p* < 0.001. VIT: vitreous; TAC: total antioxidant capacity; SCG: surrogate control group; PDRG: proliferative diabetic retinopathy group; DMEG: diabetic macular edema group.

**Figure 6 ijms-24-08227-f006:**
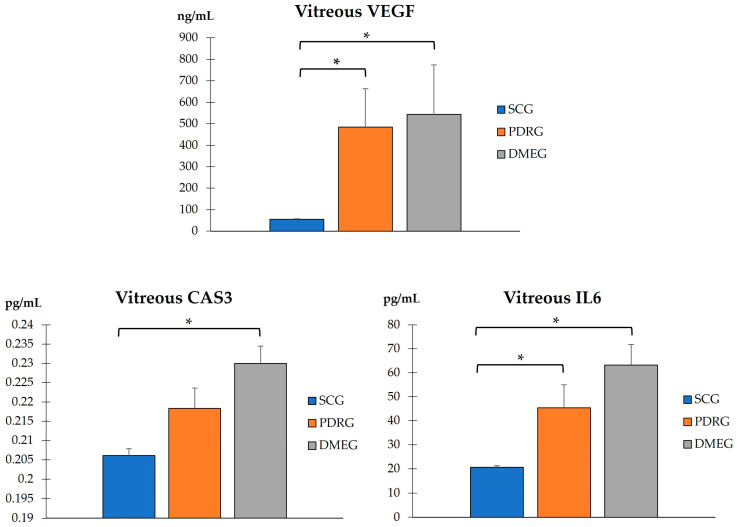
VIT levels of the pro-inflammatory IL6, pro-angiogenic VEGF, and pro-apoptotic CAS3 of the study participants. * *p* < 0.001. VIT: vitreous; IL6: interleukin 6 (pg/mL); VEGF: vascular endothelial growth factor (ng/mL); CAS3: caspase 3 (pg/mL). PDRG: proliferative diabetic retinopathy group; DMEG: diabetic macular edema group; SCG: surrogate control group.

**Figure 7 ijms-24-08227-f007:**
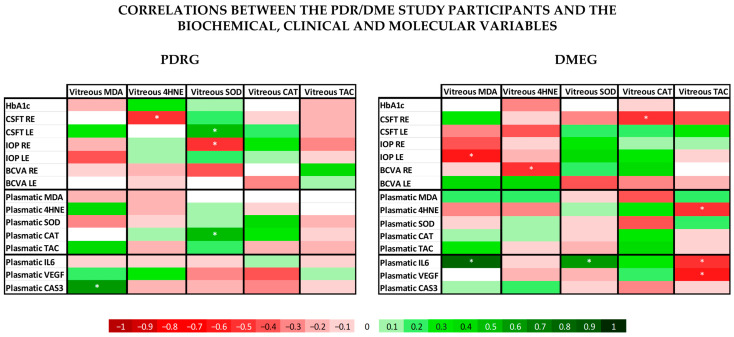
Data from the correlation analysis of the clinical, biochemical, and molecular variables. According to the bottom legend, the correlation coefficients for different continuous variables in PDRG are shown on the left side, and those for DMEG are shown on the right side. * indicates that the correlation coefficient is statistically different from 0 (*p* < 0.05). PDRG: proliferative diabetic retinopathy group; DMEG: diabetic macular edema group; SCG: surrogate control group. RE: right eye; LE: left eye; HbA1c: glycated hemoglobin; BCVA: best-corrected visual acuity; CSFT: central subfield foveal thickness in OCT; CAT: cube average thickness on OCT; IOP: intraocular pressure; PLS: plasma; VIT: vitreous.

**Figure 8 ijms-24-08227-f008:**
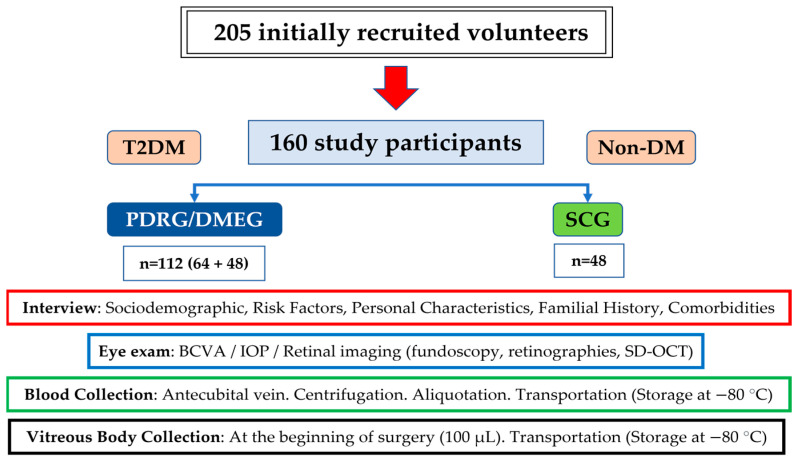
Flowchart of the recruitment characteristics and proceedings of the study.

**Table 1 ijms-24-08227-t001:** Sociodemographic data and characteristics of the study participants.

	SCG	PDRG	DMEG
Age (years)	60 ± 9	63 ± 13	61 ± 8
Sex (Males/Females)	4/12	14/15	5/10
Affected/Operated RE (%)	45	58	44
Affected/Operated LE (%)	55	42	56
DM Duration (years)	-	19 ± 6	16 ± 3

DM: diabetes mellitus; SCG: surrogate control group; PDRG: proliferative diabetic retinopathy group; DMEG: diabetic macular edema group. RE: right eye; LE: left eye. Values are the mean ± SD.

**Table 2 ijms-24-08227-t002:** Ophthalmological characteristics of the study participants.

	SCG	PDRG	DMEG
BCVA Log MAR (RE/LE)	0.08/0.09	0.48/0.54 *	0.66/0.62 *
IOP mm Hg (RE/LE)	15 ± 2/14 ± 2	16 ± 1/15 ± 2	19 ± 1/19 ± 2 *
CSFT μm (RE/LE)	256 ± 18/247 ± 23	370 ± 46/397 ± 44 *	262 ± 10/254 ± 39 *
CAT μm (RE/LE)	264 ± 18/-	284 ± 39/290 ± 36 *	395 ± 32/399 ± 42 *

BCVA: best-corrected visual acuity; LogMAR: logarithm of the minimum angle of resolution; SCG: surrogate control group; PDRG: proliferative diabetic retinopathy group; DMEG: diabetic macular edema group; IOP: intraocular pressure; RE: right eye; LE: left eye. CSFT: central subfield foveal thickness; CAT: cube average thickness on OCT. Values are the mean ± SD. * *p* < 0.001.

**Table 4 ijms-24-08227-t004:** Summary table of the main biomarkers analyzed in the study.

**DATA COMPARISON SCG vs. PDRG**								
**Biomarker Units**	**MDA μM**	**4HNE μM**	**SOD U/mL**	**CAT U/mL**	**TAC μM**	**IL6 pg/mL**	**VEGF ng/mL**	**CAS3 pg/mL**
**Biological Sample**	Plasma	Plasma	Plasma	Plasma	Plasma	Plasma	Plasma	Plasma
***p*-value**	2.70 × 10^−7^	1.17 × 10^−3^	1.42 × 10^−6^	6.45 × 10^−5^	4.16 × 10^−14^	1.39 × 10^−4^	8.89 × 10^−10^	0.36
**Biological Sample**	Vitreous Body	Vitreous Body	Vitreous Body	Vitreous Body	Vitreous Body	Vitreous Body	Vitreous Body	Vitreous Body
***p*-value**	1.06 × 10^−22^	8.96 × 10^−14^	9.01 × 10^−13^	9.67 × 10^−18^	9.74 × 10^−8^	5.72 × 10^−6^	2.5 × 10^−3^	0.126
**DATA COMPARISON SCG vs. DMEG**								
**Biomarker Units**	**MDA μM**	**4HNE μM**	**SOD U/mL**	**CAT U/mL**	**TAC μM**	**IL6 pg/mL**	**VEGF ng/mL**	**CAS3 pg/mL**
**Biological Sample**	Plasma	Plasma	Plasma	Plasma	Plasma	Plasma	Plasma	Plasma
***p*-value**	4.11 × 10^−6^	3.24 × 10^−10^	3.32 × 10^−9^	1.28 × 10^−11^	6.10 × 10^−13^	3.87 × 10^−10^	2.45 × 10^−9^	0.53
**Biological Sample**	Vitreous Body	Vitreous Body	Vitreous Body	Vitreous Body	Vitreous Body	Vitreous Body	Vitreous Body	Vitreous Body
***p*-value**	3.06 × 10^−5^	4.44 × 10^−16^	6.26 × 10^−6^	2.30 × 10^−6^	1.57 × 10^−6^	9.09 × 10^−3^	2.59 × 10^−3^	5.44 × 10^−3^
**DATA COMPARISON PDRG vs. DMEG**								
**Biomarker Units**	**MDA μM**	**4HNE μM**	**SOD U/mL**	**CAT U/mL**	**TAC μM**	**IL6 pg/mL**	**VEGF ng/mL**	**CAS3 pg/mL**
**Biological Sample**	Plasma	Plasma	Plasma	Plasma	Plasma	Plasma	Plasma	Plasma
***p*-value**	0.2455	0.3021	3.06 × 10^−5^	1.67 × 10^−9^	1.44 × 10^−4^	2.04 × 10^−6^	0.12	0.12
**Biological Sample**	Vitreous Body	Vitreous Body	Vitreous Body	Vitreous Body	Vitreous Body	Vitreous Body	Vitreous Body	Vitreous Body
***p*-value**	7.336 × 10^−4^	1.75 × 10^−4^	0.02	1.3 × 10^−4^	8.76 × 10^−6^	0.2956	0.5677	0.1354

**Table 5 ijms-24-08227-t005:** Inclusion and exclusion criteria for the study participants.

INCLUSION	EXCLUSION
Individuals aged between 40 and 80 years, inclusive.	Individuals aged younger than 40 years or older than 80 years.
Accurate diagnosis of PDR/DME for the corresponding group of T2DM participants (PDRG).	Other DM or DR type.
Non-diabetic individuals for the comparative group of participants (CG). These can include patients suffering from macular hole (MH), epiretinal membrane (EPM), or rhegmatogenous retinal detachment (RRD).	Patients experiencing other ophthalmological diseases and/or comorbidities. Patients receiving local or systemic treatment that may interfere with the study. Eye/laser surgery in the previous 12 months.
Precise and complete data of medical history.	History including any diagnoses that do not fit the study purpose.
Adequate psycho-physical status for participating in the study.	Unfeasibility of having a thorough and complete clinical history. Unable to participate.

PDR: proliferative diabetic retinopathy; DME: diabetic macular edema; T2DM: type 2 diabetes mellitus; PDRG: proliferative diabetic retinopathy group. DM: diabetes mellitus; DR: diabetic retinopathy; MH: macular hole; EPM: epiretinal membrane; RRD: rhegmatogenous retinal detachment.

## Data Availability

Data is not available to public. If interested, authors can be reached through e-mail.

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
