# Peer review of "Oxidative Stress, Inflammatory, Angiogenic, and Apoptotic molecules in Proliferative Diabetic Retinopathy and Diabetic Macular Edema Patients"

_ijms, 2023, doi:10.3390/ijms24098227_

Round 1

Reviewer 1 Report

In this manuscript, the authors evaluated the levels of some biomarkers of oxidative stress, angiogenesis, apoptosis, and inflammation in plasma or vitreous of patients with proliferative diabetic retinopathy and diabetic macular edema compared to “surrogate” controls (other vitreoretinal diseases commonly used as “controls” in this type of studies). The scope of the manuscript is interesting but the authors failed to put into practice the purpose of the study (either experimentally or in manuscript writing). The authors also failed to describe the advantages of this study compared to others already present in the literature (that quantified the same biomarkers). The manuscript has too many inconsistencies and scientific errors to be published in the present form.

 Introduction

·      Some aspects of the introduction need to be improved. The authors go into detail about the pathogenesis mechanisms of diabetic retinopathy, as well as prevalence and risk factors (lines 53-59). However, the introduction should also serve as anticipation/preparation for the manuscript's results. In this regard, the introduction failed to explain some of the problems associated with diabetic retinopathy, which could have justified the need for this study. The sentence in lines 90-92, in my opinion, is insufficient to achieve this goal.

·      The DR has no cure. Increasing experimental and clinical evidence pinpoint that INF, ANG, and OS may be involved in DR pathogenesis, independently or all together responding to a functional link in the diabetic retina [22-26].” – lines 86-88.

It's true, diabetic retinopathy (DR) is a chronic disease and has no cure. Nevertheless, there are treatments available for the management of DR (contrarily to other types of vitreoretinal diseases whose management is basically surgical). Authors should briefly mention how the disease is managed, mainly treatments that target processes that are well referred to in the manuscript, including angiogenesis (e.g., laser, anti-VEGF) and inflammation (e.g., dexamethasone). It is also worth mentioning that several clinical trials undergoing involving therapeutic strategies for diabetes/diabetic retinopathy against oxidative stress.

Moreover, the authors could have taken the opportunity (I am referring to the second sentence) to describe some already well-known facts about the correlation between inflammation, angiogenesis, and inflammation. Even if the authors prefer not to enlarge the size of the introduction, to describe so in the introduction, I think that many of these aspects should be discussed in the manuscript.

·      “Moreover, there is growing evidence that neurodegeneration may also be an independent process occurring in DR 27, 89 28].” – lines 88-90. Neurodegeneration in ocular pathologies such as diabetic retinopathy is a “hot” topic nowadays, but it has been lightly mentioned in the manuscript. Indeed, many studies have been conducted to investigate the role of chronic neuroinflammation in diabetic retinopathy by quantifying the intravitreal levels of inflammatory mediators, adhesion molecules, and neurotrophic factors.

Results

·      “After the ophthalmological examination, vitreoretinal surgery, and sampling procedures, 78 participants were definitively included for statistical processing and distributed into: 45 T2DM patients with PDR (n=29), and/or DME (n= 16) and 33 comparatives” – lines 95-98.

“Comparatives” is an odd term to use. I understand that the controls used in this work are not “really” controls because vitreous samples are removed from patients with other vitreoretinal diseases. Consider changing it to "surrogate controls" or only "controls" (it is not incorrect since the authors explain which samples are used in this group).

·      “Main causes of reduction in the number of participants (11%) were withdraw consent, insufficient image quality, vitreous body deterioration (when shipping/storing at the laboratory, or at processing), and irregular or missing data. – lines 98-100 and Table 4.

I think that this information should be grouped. Also,  here it says “Individuals under 80 years of age” (table 4) I think it should be “Individuals over 80 years of age”.  The authors should specify which “local or systemic treatment that may interfere with the study” (anti-VEGF, corticosteroid, or laser)

·      The number of patients in each group should be included in tables 1, 2, and 3.

·      Out of curiosity, what is the reason to mention the values for ophthalmological characteristics for the right and left eye separately?

·      Please, replace the “star symbol” with “*” as referred to at the end of tables 2 and 3.

·   The analysis of the levels MDA, 4HNE, SOD, and CAT should be displayed in individual graphs to adequately adjust the scale of concentration. In the case of CAT, it is impossible to distinguish the differences, even if these differences are referred to as statistically significant.

·      “The comparison of classic lipid parameters in plasma samples among study groups showed significantly higher levels of the pro-oxidants, and significantly lower levels of the antioxidant enzymes in the PDR and DME groups than in the comparatives (Figure 1).” – lines 147-150 and “A significantly lower activity of SOD, CAT, and TAC was observed in the T2DM patients with PDR (after excluding other DM types, and/or patients without PDR) and the DME patients, respect to the CG” – lines 172-174.

According to the results shown in Figure 1, these sentences are not entirely correct. Higher levels of SOD were found in the PDR group compared to controls (CG), whereas lower levels of this antioxidant enzyme were found in the DME group compared with CG.

·      Why the authors didn't also do the quantification of oxidative stress markers (at least antioxidants enzymes) in the vitreous? The high antioxidant capacity of the vitreous is one of the protective mechanisms against oxidative stress in the retina, but this capacity decreases with aging and in some pathologies. On the other hand, the levels of oxidative stress markers measured in the plasma are hardly extrapolated and correlated with the events that occur in the eye during diabetic retinopathy (DR). Although oxidative stress may be one of the pathological processes underlying RD progression, it is more likely that these differences in plasma levels (between PDR and control groups) are related to the changes that occur at a systemic level due to diabetes/hyperglycemia. Therefore, the best option would have been to quantify the same variables in the vitreous and correlate them with plasma levels.

Nevertheless, I am aware of the limited volume of vitreous extracted during the vitrectomy, and that only a limited number of assays can be performed. Nonetheless, I believe that quantifying antioxidant biomarkers and correlating vitreous and plasma levels would be more novel than quantifying VEGF (a well-established biomarker of angiogenesis and proliferation in DR) or CAS3.

·      The authors should explain why in plasma analysis (Figures 1 and 2) they compare the three study groups (CG, PDRG, DMEG) but in the vitreous analysis (Figure 3) they compare only two study groups (CG, PDRG). Furthermore, it is not clear if the PDRG in the vitreous analysis includes or not the patients with DME.

·      The concentration units are changes between VEGF and CAS3 (Figure 3). In the VEGF graph, the units should be pg/mL (and not ng/mL). According to the standards used for the preparation of the standard curve (lines 412-414, the concentration units in the CAS3 graph should be ng/mL.

·      “Comparison of CAS3 levels between groups lacked statistical significance.” – lines 163-164.

In the results section, the authors refer that there is no statistical significance when comparing the CAS3 levels but there is a symbol of statistical significance in the graph. Please, remove * do gráfico CAS3.

Discussion

·    “Likely, other authors have analyzed the oxidative and antioxidant status in the vitreous body of PDR and DME patients [31]. Also, the present data agree with previous works from our research group [32, 33]. Significantly higher values of 4HNE (a cytotoxic 4-hydroxyalkenal) have been found in the vitreous samples from PDR and DME patients, respect the comparatives. – lines 179-183

This paragraph is a little bit confusing. These parameters were only quantified in the plasma in this study, correct?

·      Another thing I found curious is that the authors further discussed the results that were not statistically significant (e.g., CAS3 levels) but not discussed the IL-6 levels. Deepening the discussion about IL-6 levels should be more interesting because this cytokine was detected in plasma and vitreous (the differences were significant in the vitreous but not in the plasma).

·    The authors need to improve the discussion of their results and compare them with other studies in the literature. In addition to the research group's papers, many of which deal with oxidative stress and related signaling, here are some examples of articles that could be used in the discussion:

o   Oxidative stress:

§  Brzović-Šarić, Vlatka, et al. "Levels of selected oxidative stress markers in the vitreous and serum of diabetic retinopathy patients." Molecular vision 21 (2015): 649.

§  Gehl, Z., et al. "Diabetes-induced oxidative stress in the vitreous humor. Redox Biol. 9, 100–103." (2016).

§  Suzuki, Yuta, et al. "Elevation of the vitreous body concentrations of oxidative stress-responsive apoptosis-inducing protein (ORAIP) in proliferative diabetic retinopathy." Graefe's Archive for Clinical and Experimental Ophthalmology 257 (2019): 1519-1525.

§  Ankamah, Emmanuel, et al. "Vitreous antioxidants, degeneration, and vitreo-retinopathy: exploring the links." Antioxidants 9.1 (2019): 7.

§  Bokhary, Kholoud, et al. "Role of oxidative stress and severity of diabetic retinopathy in type 1 and type 2 diabetes." Ophthalmic Research 64.4 (2021): 613-621.

§ Santos, Fátima Milhano, et al. "Vitreous Humor Proteome: Targeting Oxidative Stress, Inflammation, and Neurodegeneration in Vitreoretinal Diseases." Antioxidants 11.3 (2022): 505.

o   IL-6/VEGF

§  Suzuki, Yukihiko, et al. "Expression profiles of cytokines and chemokines in vitreous fluid in diabetic retinopathy and central retinal vein occlusion." Japanese journal of ophthalmology 55 (2011): 256-263.

§  Koskela, U. E., et al. "High vitreous concentration of IL-6 and IL-8, but not of adhesion molecules in relation to plasma concentrations in proliferative diabetic retinopathy." Ophthalmic research 49.2 (2013): 108-114.

§  Wang, Jiaxing, et al. "Vitreous and plasma VEGF levels as predictive factors in the progression of proliferative diabetic retinopathy after vitrectomy." PLoS One 9.10 (2014): e110531.

§  Klaassen, Ingeborg, et al. "Identification of proteins associated with clinical and pathological features of proliferative diabetic retinopathy in vitreous and fibrovascular membranes." PLoS One 12.11 (2017): e0187304.

§  Mesquita, Joana, et al. "VEGF-B levels in the vitreous of diabetic and non-diabetic patients with ocular diseases and its correlation with structural parameters." Medical Sciences 5.3 (2017): 17.

§  Tsai, Teresa, et al. "Anti-inflammatory cytokine and angiogenic factors levels in vitreous samples of diabetic retinopathy patients." PLoS One 13.3 (2018): e0194603.

§  Mesquita, Joana, et al. "Evaluation of the growth factors VEGF-a and VEGF-B in the vitreous and serum of patients with macular and retinal vascular diseases." Growth Factors 36.1-2 (2018): 48-57.

Materials and Methods

·      I missed the correlation analysis between the medical parameters (e.g., BCVA, IOP, CSFT) and the levels of the different quantified biomarkers. Curiously, the authors refer to this type of analysis in the materials and methods (lines 430-432) but no results are shown.  

·      Similarly, the authors refer that “TAC, was measured in the vitreous samples by the antioxidant assay kit (Ref: 709001, 393 Cayman Chemical Company, Ann Arbor, MI, USA)” – lines 393-397, but not no results are shown.  

·      The section “4.5 Analytical Laboratory Procedures” from “Materials and methods” appears to have been copied and pasted directly from the manufacturer’s instructions for quantification kits. The different analytical assays should be divided by section. The authors should improve the description of these methods!

Author Response

REVIEWER 1

In this manuscript, the authors evaluated the levels of some biomarkers of oxidative stress, angiogenesis, apoptosis, and inflammation in plasma or vitreous of patients with proliferative diabetic retinopathy and diabetic macular edema compared to “surrogate” controls (other vitreoretinal diseases commonly used as “controls” in this type of studies). The scope of the manuscript is interesting but the authors failed to put into practice the purpose of the study (either experimentally or in manuscript writing). The authors also failed to describe the advantages of this study compared to others already present in the literature (that quantified the same biomarkers). The manuscript has too many inconsistencies and scientific errors to be published in the present form.

The authors would like to thank The Reviewer-1 for specific important comments to our work, the positive criticism, and the time invested in the reviewing process. As a result of this, an important number of sentences have been changed, and others have been deleted. Some new references have also been enclosed in the revised version of this work. All changes have been marked in yellow in the latest version of our ms and illustrations. Thanking so much for taking the time to assess our ms, we would like to answer point by point all the concerns The Reviewer raised. We firmly recognize that this review process has been extremely useful for improving our ms.

Introduction

1) Some aspects of the introduction need to be improved. The authors go into detail about the pathogenesis mechanisms of diabetic retinopathy, as well as prevalence and risk factors (lines 53-59).

  • “The DR has no cure. Increasing experimental and clinical evidence pinpoint that INF, ANG, and OS may be involved in DR pathogenesis, independently or all together responding to a functional link in the diabetic retina [22-26].” – lines 86-88.
  • Therefore, in agreement with the pinpointed question: “Authors should briefly mention how the disease is managed, mainly treatments that target processes that are well referred to in the manuscript, including angiogenesis (e.g., laser, anti-VEGF) and inflammation (e.g., dexamethasone).

The authors thank The Reviewer for calling our attention to these important points of the Introduction section. In fact, most of this section has been rewritten and new references have been enclosed. We agree with The Reviewer that diabetic retinopathy (DR) is a chronic disease and has no cure. Nevertheless, there are some medical-laser-surgical treatments available for the management of DR (contrarily to other types of vitreoretinal diseases whose management is basically surgical), that unfortunately we did not reflect in the former ms. Regarding this topic, two new sentences and the corresponding references have been enclosed in the introduction and bibliography sections of the revised version of the ms to clarify the importance of DM early detection and management to avoid visual impairment and blindness in diabetics (please, see in page 2, the two last paragraphs).  

  • “Current diagnostic and therapeutic tools, and outstanding new perspectives, including the digital revolution for health care (telemedicine, artificial intelligence, wearable sensors, new apps, etc.), and enabling innovative medical research in risk factors, pathophysiological mechanisms, and underlying predictors, may help to significantly gain insights for diabetic eye care and management”.
  • “Up today, most used key actions for DR and DME includes stern metabolic control and occurrence of related comorbidities (hypertension blood pressure, dyslipemia, obesity, etc) (15,22-24). However, the advanced cases need laser treatment, intravitreal therapy (anti-VEGF, steroid), and vitreoretinal surgery. Concerning to prevent progression it is widely accepted that pan-retinal laser photocoagulation is useful for retinal ischemic disease, as well as the grid, sub-threshold diode micropulse laser photocoagulation for DME [25-27]. Furthermore, intravitreal injection of anti-VEGF and/or steroid treatment has been extensively used for PDR and DME [25-[29]. Vitrectomy is conventionally performed in cases of vitreous hemorrhage and/or tractional retinal detachment [29]. Even the described DR/DME treatments, an important number of patients experience complications and visual impairment. Thus far, no curative treatments for DR exists, and there is still a long trip to avoid visual impairment and blindness in diabetics”

2) However, the introduction should also serve as anticipation/preparation for the manuscript's results. In this regard, the introduction failed to explain some of the problems associated with diabetic retinopathy, which could have justified the need for this study. The sentence in lines 90-92, in my opinion, is insufficient to achieve this goal.

The Reviewer have brought up some important points about the introduction section of our work. Thank you indeed for raising this point. We have prepared a new sentence based on the above comments (please see the page 3, last three paragraphs).

  • “In our previous works, the expression of pro-oxidant (malondialdehyde: MDA; 4-hydroxinonenal: 4HNE), pro-inflammatory [interleukins: (IL) IL1b, IL6; tumor necrosis factor (TNF) a], pro-angiogenic (vascular endothelial growth factor: VEGF) and pro-apoptotic [caspase 3: CAS3; poly adenyl ribose polymerase (PARP) 1] proteins, were dangerously increased, while the levels of antioxidants [superoxido dysmutase: SOD; catalase: CAT; total antioxidant capacity: TAC; vitamin C) were decreased in DM2 patients with NPDR and PDR (NUESTROS ART). The above results strongly specify that oxidative stress, inflammation, angiogenesis, and apoptosis are involved in the pathogenesis of DR/DME. Taken all together, these findings put forward that structural/functional damage to the vascular endothelial cells, pericytes, and resident glia induces deleterious changes to the retinal neurons and finally their apoptotic death, pathogenetic hallmarks in DR/DME.
  • “Therefore, the present work aimed to summarize current knowledge on the biochemical and molecular mechanisms involved in DR and DME, by exploring the cascade of events related to oxidative stress, inflammation, angiogenesis and apoptosis pathological pathways, and the correlation of data obtained from blood and vitreous humor samples from DM2 patients, thus to validate previous results, as well as to address potential issues for precision medicine on DR/DME diagnostic and therapy, based on hallmark mechanisms of actions. Identification of more specific and sensitive biomarkers are relevant for facilitating the earlier detection of DR/DME for reducing the prevalence of visual disability”

3) It is also worth mentioning that several clinical trials undergoing involving therapeutic strategies for  diabetes/diabetic retinopathy against oxidative stress.

We greatly appreciate The Reviewer for notifying us about this matter. In fact, it has been a pivotal part of our research line in DR/DME through the past years. We have stated the important role of oxidative stress and inflammation, and decreased antioxidant defences in DR progression to PDR stage, and a variety of therapeutic strategies for the affected patients have been reported. We have also done interventional studies regarding the antioxidant and anti-inflammatory effects of natural food and nutraceutics in DR. Some references of our research group and others have been included for supporting this topic in the former and currently revised ms [please, we would like to carry you to the introduction section, (page 3, third paragraph), and the bibliography section (new references: 14,15,39-46]. This point has also been revisited in the new discussion version (pages xxx to xxx).

4) Moreover, the authors could have taken the opportunity (I am referring to the second sentence) to describe some already well-known facts about the correlation between inflammation, angiogenesis, and inflammation. Even if the authors prefer not to enlarge the size of the introduction, to describe so in the introduction, I think that many of these aspects should be discussed in the manuscript.

We agree with The Reviewer. We have carefully revisited the important topic The Reviewer has raised, and a new sentence, and some new references has been prepared for answering this important question. According to The Reviewer suggestions, we have decided to enclose a new paragraph in the discussion section.

  • Cellular interactions in the context of biochemical and metabolic dysregulation (polyol pathway, protein kinase C activation, upregulation of advanced glycation end-products, and activation of renin angiotensin system) occurs in the diabetic retina. Therefore, the described biochemical changes induce a series of events, such as oxidative stress, inflammation, angiogenesis, and apoptosis, which harm the retina. Cellular damage and loss of vascular endothelial cells and pericytes, and basement membrane thinning are important initiators of DR pathology. Cytokines increases (IL-1β, IL-6, TNF-α), and neuroprotective and neurotrophic molecules loss, also occurs in the hyperglycemic athmosphere [33,34]. From a molecular viewpoint, the persistent hyperglycemia manages the formation of reactive oxygen species (ROS) and pro-inflammatory mediators. The increased ROS activates the inflammatory mediators, while the immune response increases ROS formation in a stepwise fashion, that strongly suggest a cross-talk between these pathologic pathways [33-35,52-54]. Then, hypoxia triggers the expression of pro-angiogenic factors, being the most representative the VEGF [55].

5) “Moreover, there is growing evidence that neurodegeneration may also be an independent process occurring in DR 27, 89 28].” – lines 88-90. 

Neurodegeneration in ocular pathologies such as diabetic retinopathy is a “hot” topic nowadays, but it has been lightly mentioned in the manuscript. Indeed, many studies have been conducted to investigate the role of chronic neuroinflammation in diabetic retinopathy by quantifying the intravitreal levels of inflammatory mediators, adhesion molecules, and neurotrophic factors.

The Reviewer is right, and this is a very nice piece of advice. Thank you. Therefore, we have prepared a new paragraph and new references in this concern (please, see page 3, 2nd paragraph), according to the comments and suggestions.

.

  • Moreover, there is growing evidence that neurodegeneration may also be an independent process occurring in DR [4.5.36-39]. “Regarding the pathophysiology, increasing research have demonstrated that DR is a complex neurovascular disease in which the chronic hyperglycaemia affects both the retinal vasculature and neural tissue [39]. The neurovascular unit/neurovascular couple (NVU/NVC) concept was introduced from the CNS to the retina in the millenium[4,5] by refering to the dysfunctional inter-relationship between neuronal, glial, inmune, and vascular retinal cell phenotypes that became unable to maintain the homeostasis, the inner BBR integrity, as well as to orchestrate the blood flow responses to the high metabolic retinal demand. Epidemiological and experimental studies reported that blood flow regulation and neurovascular coupling deficiencies may exist before the presence of neural dysfunction in the diabetic retina [39]. In addition, collection of advanced glycation end products (AGEs), induction of the protein kinase C (PKC), as well as the polyol, and the hexosamine pathways have been classicaly considered pathogenic mechanisms of DR [35,36]. In this regard, oxidative stress, neuroinflammation and glutamate-dependent excitotoxicity worsen the retinal cells survival. Finally, the affected NVC of the retina lead to hypoxia and progressive neurodegeneration, neuroinflammation, gliosis, blood retinal barrier dysfunction, edema, angiogenesis and fibrosis with increasing expression levels of several molecules, such as the cytokines and pro-angiogenic factors (as the VEGF), with all the above processes being inter-connected during the DR/DME course [35.36-40]”.

Material and Methods

1) I missed the correlation analysis between the medical parameters (e.g., BCVA, IOP, CSFT) and the levels of the different quantified biomarkers. Curiously, the authors refer to this type of analysis in the materials and methods (lines 430-432) but no results are shown.  

We are extremely grateful to The Reviewer for raising this important point for our work. As we have mentioned in previous commentaries, this topic has been addressed, and new paragraphs have been already incorporated to the new version of the ms. Unfortunately, we did not included the correlation analysis in our former ms, but we have done it after receiving your comments. We hope this issue has been solved with our modifications (please, see new results section).

2) Similarly, the authors refer that “TAC, was measured in the vitreous samples by the antioxidant assay kit (Ref: 709001, 393 Cayman Chemical Company, Ann Arbor, MI, USA)” – lines 393-397, but not no results are shown.  

Again, this has been added. TAC was determined in the vitreous body samples and the results have been enclosed in the new version of the manuscript.

3) The section “4.5 Analytical Laboratory Procedures” from “Materials and methods” appears to have been copied and pasted directly from the manufacturer’s instructions for quantification kits. The different analytical assays should be divided by section. The authors should improve the description of these methods!

This is somehow true, and we apologise for the inconvenience: we based most of the parts of this section on the protocols from the commercial kits we used. However, this has been already modified, and the new version, has been much improved in this concern, according to The Reviewer suggestion. Furthermore, the statistical proceedings have also been rewritten in the current revised version of our work (as reflected on page   lines ….).

Results

1) “After the ophthalmological examination, vitreoretinal surgery, and sampling procedures,  78 participants were definitively included for statistical processing and distributed into: 45 T2DM patients with PDR (n=29), and/or DME (n= 16) and 33 comparatives” – lines 95-98.

“Comparatives” is an odd term to use. I understand that the controls used in this work are not “really” controls because vitreous samples are removed from patients with other vitreoretinal diseases. Consider changing it to "surrogate controls" or only "controls" (it is not incorrect since the authors explain which samples are used in this group).

Thank you again for outstanding help. We have considered this and thought it is a good idea to change it, as suggested. Therefore, comparatives were named surrogate controls/surrogate control group (SCG) of participants, and all changes regarding this group were marked in the revised version of the ms and illustrations. We have also reflected in the revised section the number of participants in each group according to the new data that we have added, according to The Reviewer suggestions.

2) “Main causes of reduction in the number of participants (11%) were withdraw consent, insufficient image quality, vitreous body deterioration (when shipping/storing at the laboratory, or at processing), and irregular or missing data. – lines 98-100 and Table 4.

I think that this information should be grouped.

The sentence has been rewritten according to The Reviewer comments. The new sentence can be seen on page 4 (1st paragraph) of the results section. as follows:

  • “Combined patient and sampling events caused a mild reduction (11%) in the initial number of participants, and its corresponding data were excluded of statistical processing”.

3) Also, here it says “Individuals under 80 years of age” (table 4) I think it should be “Individuals over 80 years of age”. 

This was changed. 

4) The authors should specify which “local or systemic treatment that may interfere with the study” (anti-VEGF, corticosteroid, or laser)

The authors thank again The Reviewer for pointing to these interesting aspects of our work. We would like to say The Reviewer that the inclusion/exclusion criteria reflected the appropriate participant characteristics for each study group and those points have been followed during the process of volunteer recruitment. We respectfully ask The Reviewer to take a look at the table 1.

INCLUSION

EXCLUSION

Individuals with an age between 40 and 80 years, both included

Individuals with an age younger than 40 years or older than 80 years

Accurate diagnosis of PDR/DME for the corresponding group

 of T2DM participants (PDRG)

Other DM or DR type

Non-diabetic individuals for the comparative group of participants (CG). These can include patients suffering from macular hole (MH), epiretinal membrane (EPM) or rhegmatogenous retinal detachment (RRD)

Patients experiencing other ophthalmological diseases and/or comorbidities. Patients receiving local or systemic treatment that may interfere with the study. Eye/laser surgery in the previous 6 months.

Precise and complete data at the medical history

History including any diagnosis that do not fit with the study purpose

Adequate psycho-physical status for participating in the study

Unfeasibility of having a thorough and complete clinical history. Unable to participate.

In addition, we would like to show The Reviewer that the above important issue has also been emphasized in the discussion section (study limitations) on the former page but also in the current revised form

5) The number of patients in each group should be included in tables 1, 2, and 3.

We did it as requested by The Reviewer and the changes marked in yellow in the corresponding legend to the tables 1, 2, 3, and also in the new table 4.

6) Out of curiosity, what is the reason to mention the values for ophthalmological characteristics for the right and left eye separately?

We mentioned the ophthalmological characteristics for each eye in separate because our clinical protocols are prepared in this sense. Also the data sheet for recording all the variables necessary for the studies it was designed for collecting data from each eye. That is the reason why we have decided to follow the procedure and include the results of each eye in the text and illustrations.

7) Please, replace the “star symbol” with “*” as referred to at the end of tables 2 and 3.

It was done

8) The analysis of the levels MDA, 4HNE, SOD, and CAT should be displayed in individual graphs to adequately adjust the scale of concentration. In the case of CAT, it is impossible to distinguish the differences, even if these differences are referred to as statistically significant.

All these aspects have already been reviewed and changed, as suggested. In fact, the tables and figures have been done again. We consider that this is a very adequate way of improving the manuscript. The authors thank The Reviewer for the positive criticism. 

9) “The comparison of classic lipid parameters in plasma samples among study groups showed significantly higher levels of the pro-oxidants, and significantly lower levels of the antioxidant enzymes in the PDR and DME groups than in the comparatives (Figure 1).” – lines 147-150 and “A significantly lower activity of SOD, CAT, and TAC was observed in the T2DM patients with PDR (after excluding other DM types, and/or patients without PDR) and the DME patients, respect to the CG” – lines 172-174.

According to the results shown in Figure 1, these sentences are not entirely correct. Higher levels of SOD were found in the PDR group compared to controls (CG), whereas lower levels of this antioxidant enzyme were found in the DME group compared with CG.

All data from the assayed molecules have been carefully reviewed. The tables and figures have been done again. The Reviewer suggestions have been taken into consideration.

10) Why the authors didn't also do the quantification of oxidative stress markers (at least antioxidants enzymes) in the vitreous? The high antioxidant capacity of the vitreous is one of the protective mechanisms against oxidative stress in the retina, but this capacity decreases with aging and in some pathologies. On the other hand, the levels of oxidative stress markers measured in the plasma are hardly extrapolated and correlated with the events that occur in the eye during diabetic retinopathy (DR). Although oxidative stress may be one of the pathological processes underlying RD progression, it is more likely that these differences in plasma levels (between PDR and control groups) are related to the changes that occur at a systemic level due to diabetes/hyperglycemia. Therefore, the best option would have been to quantify the same variables in the vitreous and correlate them with plasma levels.

Nevertheless, I am aware of the limited volume of vitreous extracted during the vitrectomy, and that only a limited number of assays can be performed. Nonetheless, I believe that quantifying antioxidant biomarkers and correlating vitreous and plasma levels would be more novel than quantifying VEGF (a well-established biomarker of angiogenesis and proliferation in DR) or CAS3.

First, we would like to thank The Reviewer for raising this really important point to consider. A careful reading of our former ms lead us to detect an important failure with the data reflected in the results section. The text and illustrations are shown as data from the plasma, but those pertain to the vitreous body. We apologyse for any problem caused with this failure. The text and illustrations appear conveniently in the present revised form. Anyway, answering your brilliant question, we did quantify the oxidative stress markers’ levels and those results were added when we reviewed the manuscript again. Same thing with the correlations analyses you’ve mentioned.

11) The authors should explain why in plasma analysis (Figures 1 and 2) they compare the three study groups (CG, PDRG, DMEG) but in the vitreous analysis (Figure 3) they compare only two study groups (CG, PDRG). Furthermore, it is not clear if the PDRG in the vitreous analysis includes or not the patients with DME.

We would like to thank The Reviewer for raising this point. We have failed in organizing the results in our previous ms. We apologise for this mistake. Main molecules regarding oxidative stress, inflammation, angiogenesis, and apoptosis were determined in both samples. The plasma and vitreous assayed molecules have now been conveniently enclosed in the new text and illustrations (please, see new pages   to   ).

12) The concentration units are changes between VEGF and CAS3 (Figure 3). In the VEGF graph, the units should be pg/mL (and not ng/mL). According to the standards used for the preparation of the standard curve (lines 412-414, the concentration units in the CAS3 graph should be ng/mL.

We thank The Reviewer for helping so much with this review process. That point, has been conveniently corrected.

13) “Comparison of CAS3 levels between groups lacked statistical significance.” – lines 163-164.

In the results section, the authors refer that there is no statistical significance when comparing the CAS3 levels but there is a symbol of statistical significance in the graph. Please, remove * do gráfico CAS3.

We are truly sorry for these mistakes. There must have been some type of confusion while preparing those graphics. You were right and all the issues you are highlighting have already been changed, as kindly suggested.

Discussion

1) “Likely, other authors have analysed the oxidative and antioxidant status in the vitreous body of PDR and DME patients [31]. Also, the present data agree with previous works from our research group [32, 33]. Significantly higher values of 4HNE (a cytotoxic 4-hydroxyalkenal) have been found in the vitreous samples from PDR and DME patients, respect the comparatives. – lines 179-183

This paragraph is a little bit confusing. These parameters were only quantified in the plasma in this study, correct?

Answering your question, these parameters were also quantified in the vitreous body samples, but the results were not added to the first version of the manuscript. The results have been reviewed and conveniently included in the results section.

2) Another thing I found curious is that the authors further discussed the results that were not statistically significant (e.g., CAS3 levels) but not discussed the IL-6 levels. Deepening the discussion about IL-6 levels should be more interesting because this cytokine was detected in plasma and vitreous (the differences were significant in the vitreous but not in the plasma).

You are right, and this issue has already been revised, and the sentences have been rewritten.

3) The authors need to improve the discussion of their results and compare them with other studies in the literature.

Thanking again The Reviewer for outstanding help with the evaluation of our work, we would like to answer that the discussion section has been rewritten in part, according to The Reviewer suggestions. Now, we firmly believe that the text and illustrations can be more easily followed by the readers.

In addition to the research group's papers, many of which deal with oxidative stress and related signaling, here are some examples of articles that could be used in the discussion:

o   Oxidative stress:

  • Brzović-Šarić, Vlatka, et al. "Levels of selected oxidative stress markers in the vitreous and serum of diabetic retinopathy patients." Molecular vision 21 (2015): 649.
  • Gehl, Z., et al. "Diabetes-induced oxidative stress in the vitreous humor. Redox Biol. 9, 100–103." (2016).
  • Suzuki, Yuta, et al. "Elevation of the vitreous body concentrations of oxidative stress-responsive apoptosis-inducing protein (ORAIP) in proliferative diabetic retinopathy." Graefe's Archive for Clinical and Experimental Ophthalmology 257 (2019): 1519-1525.
  • Ankamah, Emmanuel, et al. "Vitreous antioxidants, degeneration, and vitreo-retinopathy: exploring the links." Antioxidants 9.1 (2019): 7.
  • Bokhary, Kholoud, et al. "Role of oxidative stress and severity of diabetic retinopathy in type 1 and type 2 diabetes." Ophthalmic Research 64.4 (2021): 613-621.
  • Santos, Fátima Milhano, et al. "Vitreous Humor Proteome: Targeting Oxidative Stress, Inflammation, and Neurodegeneration in Vitreoretinal Diseases." Antioxidants 11.3 (2022): 505.

o   IL-6/VEGF

  • Suzuki, Yukihiko, et al. "Expression profiles of cytokines and chemokines in vitreous fluid in diabetic retinopathy and central retinal vein occlusion." Japanese journal of ophthalmology 55 (2011): 256-263.
  • Koskela, U. E., et al. "High vitreous concentration of IL-6 and IL-8, but not of adhesion molecules in relation to plasma concentrations in proliferative diabetic retinopathy." Ophthalmic research 49.2 (2013): 108-114.
  • Wang, Jiaxing, et al. "Vitreous and plasma VEGF levels as predictive factors in the progression of proliferative diabetic retinopathy after vitrectomy." PLoS One 9.10 (2014): e110531.
  • Klaassen, Ingeborg, et al. "Identification of proteins associated with clinical and pathological features of proliferative diabetic retinopathy in vitreous and fibrovascular membranes." PLoS One 12.11 (2017): e0187304.
  • Mesquita, Joana, et al. "VEGF-B levels in the vitreous of diabetic and non-diabetic patients with ocular diseases and its correlation with structural parameters." Medical Sciences 5.3 (2017): 17.
  • Tsai, Teresa, et al. "Anti-inflammatory cytokine and angiogenic factors levels in vitreous samples of diabetic retinopathy patients." PLoS One 13.3 (2018): e0194603.
  • Mesquita, Joana, et al. "Evaluation of the growth factors VEGF-a and VEGF-B in the vitreous and serum of patients with macular and retinal vascular diseases." Growth Factors 36.1-2 (2018): 48-57.

The above references have been carefully studied, and most have been incorporated to the new ms.

We would like to thank The Reviewer for having invested your time giving us these references, since you have made our reviewing tasks way easier. Your dedication to this reviewing process was astonishing, and so helpul to us, in order to acquire more knowledge in the field and to improve our skills.

Reviewer 2 Report

11.      Highlight your contributions more precisely. 

22.   Highlight the limitations of existing works discussed in the literature (related work).

  3.   Discuss more recent works to strengthen the literature survey.

44. Improve the result and discussion. 

Author Response

REVIEWER- 2

  1. Highlight your contributions more precisely. 
  2. Highlight the limitations of existing works discussed in the literature (related work).
  3. Discuss more recent works to strengthen the literature survey.
  4. Improve the result and discussion.

The authors would like to thank The Reviewer for useful comments. All the suggestions you gave us have already been considered and added to the manuscript.

  • Highlight your contributions more precisely. 

As we know, the introduction of our work have to serve as information for the study purpose. However, we realized that we totally failed to explain the most relevant problems associated with DR/DME. We thank The Reviewer for calling our attention to this pivotal point of our work. We have been working in this matter, and some new sentences and its corresponding references have been enclosed (marked in yellow) in the introduction, discussion and bibliography sections of the current version of the ms, hoping that now, the purpose of this work can be better justified and better followed by the readers.

For example:

  • “Given the relatively recent trend toward biomarker discovery and translation to the clinical practice, we aimed to explore the cascade of events related to oxidative stress, inflammation, angiogenesis and apoptosis pathological pathways, and the correlation of data obtained from blood and vitreous humor samples from PDR, DME patients, and surrogate controls”. Please see page 3, lines 147-151 of the new introduction section.
  • “In this work we intend to address potential issues for precision medicine in PDR/DME diagnostic and therapy, based on hallmark mechanisms of actions. Identification of more specific and sensitive biomarkers with a known relationship with the specific clinical endpoint, is relevant for facilitating the earlier detection of DR/DME for reducing the prevalence of visual disability”. Please, see page  12, lines 315-319 of the new discussion section.
  • “Key research in PDR/DME focuses on the molecular-genetic mechanisms (biomarkers, gene editing), advanced diagnostic techniques (deep learning authomated algorithsm for imaging, robotics), and application of stem cell appliances”.  Please, see page 13, lines 411-413 of the new discussion section.
  • ”This work aimed to revisite current knowledge and to recognize the most pressing problems in DR/DME, such as preventing and treating DR/DME which are major concerning challenges in this field. New strategies are urgently needed for preclinically detecting potential vision-threatening retinopathy and maculopathy, and prevent its progression, in diabetics”. Please, see page 13, lines 413-418 of the discussion section, as the summary.

  • Highlight the limitations of existing works discussed in the literature (related work).

We agree with The Reviewer that we have to enclose the limitations of the works that have been discusses in the corresponding section. First, we have noticed, that despite all the attempts to develop novel prognostic and diagnostic biomarkers for PDR/DME, few current trials use biomarkers other than the imaging parameters, or the HbA1c, VEGF, or IL6 as entry criteria. Additionally, the studies are extremely heterogeneous. We have been writting the most relevant limitations of the studies basically discussing the research design, sample size (representative or not of the target population and the the fulfillment of the objectives of the study), any unanswered questions, the possibility of using another method of data collection, reproducibility, feasible trasnlation  etc,  All discussed publications have been compared with ours.

  • Discuss more recent works to strengthen the literature survey.

Again, we thank The Reviewer for for warning us about this particular point. It has been done.  We have done an extensive search of the scientific liteerature. The current bibliography section includes more recent works and most important journals in this field.

  • Improve the result and discussion.

The authors would like to answer The Reviewer that the introduction, results and discussion section have been rewritten in part, according to The Reviewer suggestions. To be honest, we firmly believe that the text and illustrations (new tables and figures) have been greatly improved and in the present revised form can be more easily followed by the readers.

Reviewer 3 Report

Dear author

With the best respects, this study needs to explain more in methodology and result sections.

Author Response

REVIEWER-3

Dear author

With the best respects, this study needs to explain more in methodology and result sections.

We appreciate your feedback and your advice. It is true: the methodology and result sections can be improved, and we have worked in that direction.

Regarding methodology, most of the former parts we wrote have been now changed. Please see the new ms, the material and methods section (pages )  and the results section (pages )

Thank you again for your help.

Reviewer 4 Report

In this study the authors have carried out a clinical study comparing a number of factors in controls, proliferative diabetic and macular edema patients.  The cohort sizes are reasonable but not very large for a study of this type.

In the introduction the authors describe DR as a microangiogenic disease.  They need to discuss the views expressed in multiple papers that vascular involvement is a late stage effect following neurodegenerative and inflammatory processes.

The authors measure a series of vitreous and serum factors.  All of the findings have been described in many previous papers and there is nothing unique in the findings.  For example the review article by López-Contreras et al., 2020, Antioxidants, 9:891, discusses all the factors described in the present manuscript and provides references to the extensive literature.

There is also no attempt to relate the various findings to each other or to overall mechanisms.

A common problem with studies such as this is that they are a single time point.  How do these factors change with progression of the disease?  Were there any treatments given to the PDR and DME groups and how did this change any of the factors measured (at lest the serum factors)?

One minor point.  In Figure one SOD looks higher than the control group although the text says it is lower.

Author Response

REVIEWER-4

The first sentence in the abstract isn’t a complete sentence. I am not sure what the authors are trying to communicate. There are many other similar points of confusion that warrant a significant grammatical review and language edit for the manuscript. Otherwise, the manuscript was well organized and presented.

We are very grateful for the time you invested in this review and for the comments you gave us to improve our manuscript. The first sentence of the former abstract has been rewritten. We have sent our current revised ms and illustrations to the English Editing System of the Editorial, as kindly suggested by The Reviewer.

The abstract has been rewritten and the new version is:

Abstract: To evaluate molecules involving oxidative stress, inflammation, angiogenesis, and apoptosis, and to discern which are more likely implicated in proliferative diabetic retinopathy (PDR) and diabetic macular edema (DME), and to investigate the correlation between them in vitreous body and plasma, as well as data obtained from the ophthalmological examination. Suitable type 2 diabetic (T2DM) patients with PDR/DME (PDRG/DMEG; n=112) and non-DM subjects as the surrogate controls (CG; n=48) were selected according to the inclusion/exclusion criteria and programmed for vitrectomy either because their PDR or by having macular hole (MH)/epiretinal membrane (ERM)/rhegmatogenous retinal detachment. Blood samples were collected and processed to determine the glycemic profile, total cholesterol, C reactive protein, as well as the malondialdehyde (MDA), 4-hydroxynonenal (4HNE), superoxide dismutase (SOD), catalase (CAT) levels, and the total antioxidant capacity (TAC). Also, the interleukin 6 (IL6), vascular endothelial growth factor (VEGF) and caspase 3 (CAS3) were assayed. Vitreous body samples were collected and processed to measure the expression levels of all the above molecules. Statistical analyses were done using R Core Team (2022) program, including the group comparisons and correlation analyses. When comparing with the surrogate controls, our statistically significant findings support the presence of molecules involved in oxidative stress, inflammation, angiogenesis, and apoptosis in plasma (PLS) and vitreous body (VIT) samples of the PDRG/DMEG. In the PDRG the PLS antioxidant load decreased (p<0.001) whereas the proangiogenic increased (p<0.001), whereas the intravitreal pro-oxidants risen (p<0.01), and the antioxidants declined (p<0.001). For the DMEG, the PLS pro-oxidants and pro-inflammatory molecules augmented (p<0.001) and the antioxidant capacity diminished (p<0.001), and the VIT pro-oxidants increased (p<0.001) and the antioxidant charge decreased. Furthermore, the data correlation between the PDRG vs the DMEG in both samples strongly pointed to the most significant variables for PDR: the PLS -VEGF, and the VIT -MDA, -SOD and -CAT, whilst for DME: the PLS -4HNE, -VEGF and -IL6 and the VIT -TAC, indicating that these molecules display a robust correlation with PDR/DME variables. Our integrative data of T2DM patients, pointed to the above potential biomarkers as the most valuables for detecting early molecular changes of DR/DME, to identify patients at high risk of progression as well as to monitoring therapeutic outcomes in the diabetic retina.

Below are specific questions or suggestions for improvement: 

In the results, what was irregular about the data that was not used? The other reasons seem reasonable, but stating that the data was irregular seems inconsistent with best practices for data management and exclusion and inconsistent with the language used in the discussion section.

That is a very important question. Firstly, we have carefully revised all data obtained from our study participants. We have to say The Reviewer that our study also involved the meticulous collection of data from subjects based on consensus-based clinical registration forms and national biospecimen collection guidelines. The collected data were used as optimal conditions for significant viochemical and molecular screening by controlling the clinical variables as much as possible. Therefore, even though a small number of subjects were used for the study, markers with high diagnostic value could be screened. According to the study design, some of the data registered were excluded of the present work. Some of these were incomplete, or confusing, but others were unintentionally forgotten for statistical processing. We have decided to focus on the main purpose of the study by using the main sociodemographic, ophthalmological data (including the OCT imaging parameters), as well as the results of the assayed molecules regarding oxidative stress, inflammation, angiogenesis, and apoptosis that were obtained from both the plasma and vitreous body samples of the PDR and DME study participants. We want to say The Reviewer that we did not included some plasma and vitreous data, by mistake. Fortunately, as a result of this revision, we have done an extensive editing of all data, and those not previously reflected have been incorporated to the new version of our text and illustrations. The introduction, results, discussion and material and methods sections have been rewritten and all the adjustments have been marked in yellow. An important number of new references has been also enclosed. We sincerely hope that in the present revised form our work can be ready for being accepted for publication. Thank you so much for raising the above questions.

Its not surprising that malondialdehyde is increased as it has been shown to be associated with DM. IS the level increase similar to other published studies? Similar questions should be asked about the other factors.

Thanks for the suggestion. In this sense, we have to say The Reviewer that in general, yes. But it is difficult to compare such heterogeneous studies. Important differences have been detected between the studies, basically the research design, sample size (representative or not of the target population), and some unanswered questions. Also, the biological sample used, the laboratory protocols utilized, the method of data collection,  etc,  All discussed publications have been compared with ours. Some of them are very different and others really similar regarding the oxidative stress biomarkers.

 I am not sure if the authors can draw conclusions about their measured biomarkers and their purported cause/effect of DME or DR. They are associated yes, but a single snapshot in time is not enough to conclude a cause/effect relationship.

Thank you for giving us the opportunity of speaking about this issue. We would like to respond The Reviewer that we have design our research work for the very advanced DR stage (PDR) and one of the most outstanding diabetic complications, the DME. As you know, those are the most challenging eye processes in DM course. It is very difficult to get suitable patients for both precise study groups. Furthermore, the biological samples collection has to be done at the beginning of the vitrectomy. When designing the present work, we did not think on using more than one timepoint. Because the particular characteristic of the PDR and DME patients, influences the course of the research study. It is difficult to get the signed informed consent. It is also difficult to maintain the appropriate number of participants from baseline to end-of-study. We know the described difficulties because in our previous works with NDPR and PDR (Verdejo et al., 1999; Mancino et al., 2011; Pinazo-Durán et al., 2014; Roig-Revert et al., 2015; Sanz-González et al., 2021; Fragiotta et al., 2023) we have fighting against the limitations related to this concern, by stablishing strategic contingencies. In this context, we decided to use only one timepoint, as described in the text, as the choice for outcome measures (clinical endpoints, and indirect measures, as the determination of retinal imaging, and molecules involved in main pathologic mechanisms for PDR/DME). 

Nevertheless, we are aware that this issue should be assessed, again, in future works. Thank you very much The Reviewer for this interesting viewpoint.

Why were macular hole (MH), epiretinal membrane (EPM) or regmatogenous retinal detachment (RRD) patients used for the control group? What is the breakdown of each condition? How many patients had each? Why were other conditions excluded?

These questions are quite important, and we are going to try to answer them the best way we can. Firstly, it is very important to choose the better controls for each research work. In our case, we decided to include macular hole (MH), epiretinal membrane (EPM) or rhegmatogenous retinal detachment (RRD) patients because of the following points: 1) As we have to select a comparative control group of participants, we have to do it with the same technique to collect the plasma and the vitreous body as the experimental groups. In this case, we should remember than the unique way of collecting the vitreous body, ethically from healthy individuals, is by programmed vitrectomy in the course of vitreoretinal disorders [this research was conducted according to the ethical standards for human experimentation and the World Medical Association (WMA) Declaration of Helsinki].  Therefore, it is almost impossible to achieve an appropriate sample size. 2) the participants have to sign the informed consent, and we have to precisely explain the way of vitreous body collection and the importance of this sample for research. 3) the macular hole, epiretinal membrane or rhegmatogenous retinal detachment patients are considered to have significant lower values of oxidative stress, inflammation biomarkers than other vitreoretinal pathologies, such as proliferative vitreoretinopathy or diabetic retinopathy.

Answering another question from The Reviewer, the anteroposterior traction was thought to cause direct formation of a macular hole. Subsequently, degenerative issues of the inner retinal layers, and focal anteroposterior traction mechanisms have also been proposed (Smiddy WE and Flynn HW. Pathogenesis of macular holes and therapeutic implications. Am J Ophthalmol 2004; 137(3), 525-537). Epiretinal membranes, are characterized by fibrocellular proliferations at the vitreoretinal interface, above the internal limiting membrane, which can cover partly or totally the fovea centralis. It is thought that tractional stress caused by the epiretinal membranes can induce changes in retinal thickness, formation of lamellar and full-thickness macular holes, and/or alterations of the outer foveal region. Additionally, breakdown of the retinal pigment epithelium and disruption and leakage from the retinal capillary system have also been discussed in the ethiopathogenic mechanisms of this process (Brinkmann et al., Epiretinal membrane surgery outcome in eyes with abnormalities of the central bouquet. Int J Retin Vitr 2021;7:7). Rhegmatogenous retinal detachment is a complex process that usually results from inherited and/or age-related changes in the vitreous structure and vitreoretinal adhesion, which highly predisposes to retinal break formation initiating separation of the neural retina from the underlying retinal pigment epithelium layer. It has also been reported a significant damage to the blood aqueous barrier associated with the disease (Kwon et al., Retinal detachment and proliferative vitreoretinopathy. Dev Ophthalmol. 2015, 55:154-62). Furthermore, it has been shown that the vitreous levels of proinflammatory cytokines and proangiogenic factors were significantly higher in eyes with proliferative vitreoretinopathy and diabetic proliferative retinopathy than in eyes with non-complicated rhegmatogenous retinal detachment, epiretinal membranes or macular holes, as well as in the control subjects (Takahashi et al., Profiles of Inflammatory Cytokines in the Vitreous Fluid from Patients with Rhegmatogenous Retinal Detachment and Their Correlations with Clinical Features. Biomed Res Int. 2016;2016:4256183).

And the last two points raised by The Reviewer has been answered in page 4,  lines  170-172  of the results section in the new version of our text.

  • “Constituting the SCG of non-diabetic patients were the non-complicated macular hole (n=12), epiretinal membranes (n=15) and rhegmatogenous retinal detachment (n=21) participants”.

We excluded other possibilities (uveitis, proliferative vitreoretinopathy, exudative retinal detachment) because the above disorders lacked precise conditions useful for being utilized as controls (these diseases have demonstrated to express significantly higher values of oxidative stress and inflammation biomarkers in plasma, tears, aqueous humor, and vitreous samples of the affected patients)

Many, if not all, of the biomarkers noted in this study have published associations with DE,DME, DR, etc. What makes this study unique? The authors should appropriately reference and attribute these previous studies and discuss how their data fits within this body of work.

That is a good point raised by The Reviewer. We have previously reportethe increased expression levels of proinflammatory, pro-oxidant, and proangiogenic biomarkers in plasma, tear film, aqueous humor and vitreous samples of diabetic retinopathy (NPDR / PDR) and other eye conditions. We have introduced two new sentences that may help clarifying this important question in the introduction section of the new version of the ms:

  • “Given the relatively recent trend toward biomarker discovery and translation to the clinical practice, we aimed to explore the cascade of events related to oxidative stress, inflammation, angiogenesis and apoptosis pathological pathways, and the correlation of data obtained from blood and vitreous humor samples from PDR, DME patients, and surrogate controls”.
  • “Therefore, the present work aimed to extend current knowledge on the biochemical and molecular mechanisms involved in DR and DME, by exploring the cascade of events related to oxidative stress, inflammation, angiogenesis and apoptosis pathological path-ways, and the correlation of data obtained from blood and vitreous humour samples from DM2 patients, thus to validate previous results, as well as to address potential issues for precision medicine on DR/DME diagnostic and therapy, based on hallmark mechanisms of actions. Identification of more specific and sensitive biomarkers are relevant for facilitating the earlier detection of DR/DME for reducing the prevalence of visual disability”.

We think that this work described the biomarkers that potentially can be used in the preclinical stages of retinopathy, to better eye and vision care in diabetics.       

Reviewer 5 Report

The first sentence in the abstract isn’t a complete sentence. I am not sure what the authors are trying to communicate. There are many other similar points of confusion that warrant a significant grammatical review and language edit for the manuscript. Otherwise, the manuscript was well organized and presented.

 Below are specific questions or suggestions for improvement: 

In the results, what was irregular about the data that was not used? The other reasons seem reasonable, but stating that the data was irregular seems inconsistent with best practices for data management and exclusion and inconsistent with the language used in the discussion section.

Its not surprising that malondialdehyde is increased as it has been shown to be associated with DM. IS the level increase similar to other published studies? Similar questions should be asked about the other factors.

 I am not sure if the authors can draw conclusions about their measured biomarkers and their purported cause/effect of DME or DR. They are associated yes, but a single snapshot in time is not enough to conclude a cause/effect relationship.

 Why were macular hole (MH), epiretinal membrane (EPM) or regmatogenous retinal detachment (RRD) patients used for the control group? What is the breakdown of each condition? How many patients had each? Why were other conditions excluded?

Many, if not all, of the biomarkers noted in this study have published associations with DE,DME, DR, etc. What makes this study unique? The authors should appropriately reference and attribute these previous studies and discuss how their data fits within this body of work.

Author Response

REVIEWER-5

In this study the authors have carried out a clinical study comparing a number of factors in controls, proliferative diabetic and macular edema patients.  The cohort sizes are reasonable but not very large for a study of this type.

We truly appreciate the useful commentaries of The Reviewer. First point can be answered by two main facts: 1) The vitreous samples have to be surgically collected and this need negatively influences the ease of selection of participants and limits the sample size. 2) The majority of research articles have a similar sample size to ours, probably for the difficulties in maintaining the study participants from baseline to end-of-study.

We carefully revised the text to address The Reviewer concerns, that have conducted to specific changes through the new version of the ms. Our answers are listed below.

In the introduction the authors describe DR as a microangiogenic disease. They need to discuss the views expressed in multiple papers that vascular involvement is a late stage effect following neurodegenerative and inflammatory processes.

This is a really interesting view, and a great issue to discuss. However, we have specifically selected patients with PDR and DME for this characteristic of being advanced stages of the disease with bad prognostic, in order to find biomarkers that potentially can be preclinically used to identify patients at high risk of disease progression. We have been working in this matter, and some new sentences and its corresponding references have been enclosed (marked in yellow) in the introduction, discussion and bibliography sections of the current version of the ms, hoping that now, the purpose of this work can be better justified and better followed by the readers.

For example:

  • “Given the relatively recent trend toward biomarker discovery and translation to the clinical practice, we aimed to explore the cascade of events related to oxidative stress, inflammation, angiogenesis and apoptosis pathological pathways, and the correlation of data obtained from blood and vitreous humor samples from PDR, DME patients, and surrogate controls”. Please see page 3, lines 147-151 of the new introduction section.
  • “In this work we intend to address potential issues for precision medicine in PDR/DME diagnostic and therapy, based on hallmark mechanisms of actions. Identification of more specific and sensitive biomarkers with a known relationship with the specific clinical endpoint, is relevant for facilitating the earlier detection of DR/DME for reducing the prevalence of visual disability”. Please, see page  12, lines 315-319 of the new discussion section.
  • “Key research in PDR/DME focuses on the molecular-genetic mechanisms (biomarkers, gene editing), advanced diagnostic techniques (deep learning authomated algorithsm for imaging, robotics), and application of stem cell appliances”.  Please, see page 13, lines 411-413 of the new discussion section.
  • ”This work aimed to revisite current knowledge and to recognize the most pressing problems in DR/DME, such as preventing and treating DR/DME which are major concerning challenges in this field. New strategies are urgently needed for preclinically detecting potential vision-threatening retinopathy and maculopathy, and prevent its progression, in diabetics”. Please, see page 13, lines 413-418 of the discussion section, as the summary.

Moreover, the neurodegeneration evidence has been taken into consideration, as suggested by The Reviewer. Some sentences were included in the former version of our work:

  • “Regarding the pathophysiology, increasing research have demonstrated that DR is a complex neurovascular disease in which the chronic hyperglycaemia affects both the retinal vasculature and neural tissue [39]. The neurovascular unit/neurovascular couple (NVU/NVC) concept was introduced from the CNS to the retina in the millenium[4,5] by refering to the dysfunctional inter-relationship between neuronal, glial, inmune, and vascular retinal cell phenotypes that became unable to maintain the homeostasis, the inner BBR integrity, as well as to orchestrate the blood flow responses to the high metabolic retinal demand. Epidemiological and experimental studies reported that blood flow regulation and neurovascular coupling deficiencies may exist before the presence of neural dysfunction in the diabetic retina [39]. In addition, collection of advanced glycation end products (AGEs), induction of the protein kinase C (PKC), as well as the polyol, and the hexosamine pathways have been classicaly considered pathogenic mechanisms of DR [35,36]. In this regard, oxidative stress, neuroinflammation and glutamate-dependent excitotoxicity worsen the retinal cells survival. Finally, the affected NVC of the retina lead to hypoxia and progressive neurodegeneration, neuroinflammation, gliosis, blood retinal barrier dysfunction, edema, angiogenesis and fibrosis with increasing expression levels of several molecules, such as the cytokines and pro-angiogenic factors (as the VEGF), with all the above processes being inter-connected during the DR/DME course [35.36-40]”.

The authors measure a series of vitreous and serum factors.  All of the findings have been described in many previous papers and there is nothing unique in the findings. For example the review article by López-Contreras et al., 2020, Antioxidants, 9:891, discusses all the factors described in the present manuscript and provides references to the extensive literature.

There is also no attempt to relate the various findings to each other or to overall mechanisms.

The authors thank The Reviewer for these suggestions. What The Reviewer is describing regarding our former ms is true, but the majority of publications are “Reviews”. We have already conducted the needed changes to include the correlation analyses in the text and illustrations, that are located on the results section (marked in yellow) of the new version of our work. With this new data, we add consistency to our results and the described biomarker fingerprint in plasma and vitreous of patients with PDR and DME are now more interesting. A new sentence has been enclosed in the new discussion of the current vs.

  • “Therefore, based on our results provided by the correlation analyses, plasma VEGF and intravitreal MDA and 4 HNE has been selected as biomarker candidates for distinguishing PDR cases, whereas plasma 4HNE, IL6 and VEGF and intravitreal TAC have been chosen as potential biomarkers for distinguishing DME cases”.

A common problem with studies such as this is that they are a single time point.  How do these factors change with progression of the disease? Were there any treatments given to the PDR and DME groups and how did this change any of the factors measured (at lest the serum factors)?

These questions are impressive and thought-inducing from a scientifically point of view. It is true: measuring the level of these molecules once only gives us a narrow view of what’s happening with our patients. However, the selected stages of disease that we introduced in the present work are extreme. The most common course of the PDR and DME is toward visual impairment and blindness, because DR has no cure. Because of this, the expected changes in the factors measured (plasma and vitreous) is to get worse. Nevertheless, it would be really stimulating to conduct a future study where we can follow the effect that a particular treatment may induce in the variation of these molecules’ levels. However, we would like to inform The Reviewer that in previous works of our research line in different stages of severity of DR, we addressed the suggested points, as reflected in the following references: Verdejo et al., 1999; Mancino et al., 2011; Pinazo-Durán et al., 2014; Roig-Revert et al., 2015; Sanz-González et al., 2021; Fragiotta et al., 2023.

We thank The Reviewer for raising this important point. In fact, our research group is now interested and involved in neurodegeneration and neuroprotection research in DR. The Reviewer suggestion will be addressed in our foreseeable future work.

One minor point.  In Figure one SOD looks higher than the control group although the text says it is lower.

We thank The Reviewer for calling our attention to this point. It has been changed according to this suggestion. Thank you so much for outstanding help.

Reviewer 6 Report

It is indicated in the file.

Author Response

REVIEWER-6

It is indicated in the file.

We carefully revised the text to address your concerns.

We have rewritten the introduction, results, discusión and material and methods section.

We have also made new all figures and tables. We have added a significant number of references and delete others.

We sincerely hope that our work is now clearer.

We would like to thank The Reviewer for helping so much to improve our work.

Round 2

Reviewer 2 Report

The contributions of the paper are satisfactory. However, authors are advised to do the following modifications.

                  1. Highlight the limitations of existing works discussed in the literature

       2.      Add one section to discuss the future research directions, as presented in the following papers by giving proper citations of these papers.  

[1]   “A Systematic Review on Recent Advancements in Deep and Machine Learning based Detection and Classification of Acute Lymphoblastic Leukemia,” IEEE Access, 2022.

[2]   “A review of automated methods for the detection of sickle cell disease”, IEEE Reviews in Biomedical Engineering, vol. 13, pp. 309-324, 2020.

[3] "A deep learning ensemble approach for diabetic retinopathy detection." Ieee Access 7 (2019): 150530-150539.

Reviewer 5 Report

The manuscript has been significantly improved. The authors have done a good job of addressing noted concerns.